


**Aethalometer multiple scattering correction $C_{ref}$ for mineral dust aerosols**

Claudia Di Biagio[1], Paola Formenti[1], Mathieu Cazaunau[1], Edouard Pangui[1], Nicholas Marchand[2], and

Jean-François Doussin[1]

[1] Laboratoire Interuniversitaire des Systèmes Atmosphériques (LISA), UMR 7583, CNRS, Université Paris Est

Créteil et Université Paris Diderot, Institut Pierre et Simon Laplace, Créteil, France

[2] Aix Marseille Université, CNRS, LCE, Marseille, France

Correspondence to: C. Di Biagio (cldibiagio@gmail.com) and P. Formenti (paola.formenti@lisa.u-pec.fr)

**Abstract**

In this study we provide a first estimate of the aethalometer multiple scattering correction $C_{ref}$ for

mineral dust aerosols. The $C_{ref}$ at 450 and 660 nm was obtained from the direct comparison of

aethalometer data (Magee Sci. AE31) with the absorption coefficient calculated as the difference

between the extinction and scattering coefficients measured by a CAPS PMex and a nephelometer at

450 nm and the absorption coefficient from a MAAP (Multi-Angle Absorption Photometer) at 660 nm.

Measurements were performed on seven dust aerosol samples generated in the laboratory by the

mechanical shaking of natural parent soils issued from different source regions worldwide. The single

scattering albedo (SSA) at 450 and 660 nm and the size distribution of the aerosols were also

measured.

$C_{ref}$ for mineral dust varies between 1.81 and 2.56 for a SSA of 0.85–0.96 at 450 nm and between

1.75 and 2.28 for a SSA of 0.98–0.99 at 660 nm. The calculated mean $C_{ref}$ for dust is 2.09 (± 0.22) at

450 nm and 1.92 (± 0.17) at 660 nm. With this new $C_{ref}$ the dust absorption coefficient by

aethalometer is about 2% (450 nm) and 11% (660 nm) higher than that obtained by using $C_{ref}$=2.14,

usually assumed in the literature. This difference induces up to 3% change in the dust SSA. The $C_{ref}$

seems independent of the particle fine and coarse size fractions, and so the obtained $C_{ref}$ can be

applied to dust both close to sources and following transport. Additional experiments performed with

pure kaolinite mineral and polluted ambient aerosols indicate a $C_{ref}$ of 2.49 (± 0.02) and 2.32 (± 0.01)

at 450 and 660 nm (SSA=0.96–0.97) for kaolinite, and a $C_{ref}$ of 2.32 (± 0.36) at 450 nm and 2.32 (±

0.35) at 660 nm for pollution aerosols (SSA=0.62–0.87 at 450 nm and 0.42–0.76 at 660 nm).



## 1. Introduction

Abundant and widespread in the atmosphere, mineral dust strongly contributes to the global and regional direct radiative effect and climate forcing (Highwood and Ryder, 2014; Miller et al., 2014). Mineral dust interacts through processes of scattering and absorption with both incoming shortwave radiation and outgoing terrestrial longwave radiation (Sokolik et al., 1999). As for today, the evaluation of the direct effect of mineral dust and its climate implications is still limited by the knowledge of the intensity of the dust absorption in the shortwave spectral range (Miller et al., 2004; Balkanski et al., 2007; Solmon et al., 2008; Jin et al., 2016), represented by the light absorption coefficient ($\beta_{abs}$, units of $Mm^{-1}$). The absorption coefficient of mineral dust accounts for less than ~10-20% of its total shortwave extinction, where it shows a pronounced spectral variation (Cattrall et al., 2003; Redmond et al., 2010). The highest dust absorption occurs in the UV-VIS region of the spectrum, while it levels off to null values towards the near IR (Caponi et al., 2017). As a result, its single scattering albedo (SSA), i.e. the ratio of the aerosol scattering ($\beta_{sca}$) to extinction ($\beta_{ext}=\beta_{sca}+\beta_{abs}$) coefficient, increases from values of ~0.80-0.90 at 370 nm to values of ~0.95-0.99 at 950 nm (e.g., Schladitz et al., 2009; Redmond et al., 2010; Formenti et al., 2011; Ryder et al., 2013).

Given its relatively high SSA, mineral dust can be considered as weakly absorbing in the shortwave. This is particularly true if compared to other aerosol species as soot for which the SSA in the visible may be as low as 0.2 (Bergstrom et al., 2007). Nonetheless, because of its elevated atmospheric concentration (~100÷100000 $\mu g\ m^{-3}$ close to sources and ~0.1÷100 $\mu g\ m^{-3}$ after mid– to intercontinental transport; e.g., Goudie and Middleton, 2006; Kandler et al., 2009; Querol et al., 2009; Denjean et al., 2016a), light absorption by mineral dust can be comparable to that of soot both at regional and global scale (Reddy et al., 2005; Caponi et al., 2017). Under very intense dust episodes, dust may absorb up to ~150 $Wm^{-2}$ of incoming solar radiation (Slingo et al., 2006; di Sarra et al., 2011), inducing a remarkable warming of the atmospheric layer. This strong warming can alter the atmospheric structure and stability (Heinold et al., 2008), with a possible influence on the atmospheric dynamics and meteorological fields (Pérez et al., 2006). By its direct shortwave effect dust also affects the position of the Inter Tropical Convergence Zone, which in turn influences the Western African Monsoon and modifies the pattern and intensity of rainfall over Northern Africa and the Sahel (Yoshioka et al., 2007). Nonetheless, the extent of the dust effect and its implications critically depend on the exact amount of absorbed shortwave radiation. Solmon et al. (2008), for example, showed that a small change (5%) in the shortwave SSA of dust may modify the effect of dust on the Western African Monsoon, moving from a reduction to an increase of precipitation over the Sahel.

The accurate estimation of the dust absorption over the whole shortwave range is therefore necessary to properly assess its direct radiative effect and climate implications. One instrument measuring the aerosol-light absorption from the UV to near IR range is the aethalometer (Magee Sci. AE31 model, Hansen et al., 1984; Arnott et al., 2005), operating at seven wavelengths in the 370–950 nm range. The aethalometer is used to measure the black carbon mass concentration but the spectral absorption by aerosols can be also calculated. Given its large spectral interval, the aethalometer has been used in the past to investigate the spectral dependence of dust absorption (Fialho et al., 2005;



Formenti et al., 2011), as well as the absorption by many aerosol types in different environments
(Sandradewi et al., 2008; Segura et al., 2014; Di Biagio et al., 2016; Backman et al., 2016).
The working principle of the aethalometer, a filter-based instrument, consists in measuring the
attenuation through an aerosol-laden quartz filter according to the Beer-Lambert law, used then to
derive the spectral attenuation coefficient ($\beta_{ATT}$) of the deposited particles (Hansen et al., 1984). The
"true" spectral aerosol absorption coefficient ($\beta_{abs}$) is proportional but lower than $\beta_{ATT}$ (Weingartner et
al., 2003; Collaud Coen et al., 2010; hereinafter referred as W2003 and C2010), because $\beta_{ATT}$ is
enhanced by (i) aerosol scattering towards directions different from that of the detector (scattering
effect); (ii) gradual accumulation of absorbing particles on the loaded filter, thus reducing the optical
path (shadowing effect); (iii) multiple scattering of the light beam by the filter fibres, increasing the
optical path (multiple scattering effect).
Empirical formulations of the scattering and shadowing effects are available in the literature and
permit the correction of aethalometer data for these artefacts (W2003; Arnott et al., 2005; Schmid et
al., 2006; Virkkula et al., 2007; C2010). The correction of the multiple scattering effect, however
requires the knowledge of a correction factor $C_{ref}$, which needs to be directly estimated by comparison
of aethalometer data against reference absorption measurements (W2003; C2010).
Currently data for $C_{ref}$ are available for soot particles ($C_{ref}$=2.1-2.2 at 660 nm, W2003), internally and
externally mixed soot particles and organic material ($C_{ref}$=2.3-3.9, W2003), and ambient aerosols
collected in Europe and Amazonia ($C_{ref}$=2.6-4.8, C2010; $C_{ref}$=4.9÷6.3, Saturno et al., 2016) and in the
Arctic ($C_{ref}$=3.1, Backman et al., 2016). The value most often used in the literature is 2.14 (± 0.21),
assumed as wavelength-independent (e.g., Sandradewi et al., 2008; Formenti et al., 2011; Di Biagio
2016), which corresponds to the mean of observations at 660 nm for soot aerosols (W2003). Both
W2003 and C2010, however, found a dependence of $C_{ref}$ on the aerosol single scattering albedo, with
$C_{ref}$ decreasing for increasing SSA. So, the value of 2.14 obtained for highly absorbing soot (SSA~0.2
in the visible) may not be appropriate for weakly absorbing mineral dust.
Henceforth, in this work we present the experimental estimate of an optimized $C_{ref}$ for mineral dust
aerosols at 450 and 660 nm obtained from a laboratory-based intercomparison study. Experiments
were conducted on seven dust aerosol samples generated by the mechanical shaking of natural
parent soils. Control experiments on pure kaolinite mineral, ambient aerosols sampled in the polluted
environment of the suburbs of Paris, and purely scattering ammonium sulfate, were also performed to
investigate the dependence of $C_{ref}$ on the aerosol single scattering albedo.

**2. Experimental set-up**
The experimental set-up used for the intercomparison study is shown in **Fig. 1**. The following
measurements were performed from a 8-port glass manifold (~1 L volume):
-   the absorption coefficient ($\beta_{abs}$) by a 7-wavelentgth aethalometer (Magee Sci., model AE31
working at 370, 470, 520, 590, 660, 880, 950 nm; flowrate 8 L min$^{-1}$, 2-min resolution) and a MAAP
(Multi-Angle Absorption Photometer, Thermo Sci., model 5012 working at 670 nm; flowrate 8 L



min$^{-1}$, 1-min resolution). Unlike the aethalometer, the MAAP measures the transmitted light from
the aerosol-laden filter and also the backscattered light at two angles (135° and 165°) (Petzold et
al., 2004). Backscattering measurements are used to constrain the scattering fraction of the
measured attenuation that would erroneously be interpreted as absorption. The aerosol absorption
coefficient for the MAAP is obtained from a radiative transfer scheme taking into account the
multiple scattering in the filter and the scattering effect, without requiring any further adjustment
(Petzold and Schönlinner, 2004). The MAAP is commonly assumed to provide the most reliable
direct estimate of the aerosol absorption coefficient at a single wavelength (Andreae and Gelècser
2006). In this study we assume for the MAAP the manufacturer's reported wavelength of 670 nm,
even if Müller et al. (2011) measured for this instrument a wavelength of 637 nm;
-    the scattering coefficient ($\beta_{sca}$) in the 7-170° angular range by a 3-wavelentgth nephelometer (TSI

Inc., model 3563 working at 450, 550 and 700 nm; flowrate 18 L min$^{-1}$, 1-s resolution);

-    the extinction coefficient ($\beta_{ext}$) by two Cavity Attenuated Phase Shift Extinction analyzers (CAPS

PMex by Aerodyne, one working at 450 nm and the other at 630 nm; flowrate 0.85 L min$^{-1}$, 1-s

resolution);

-    the particle number size distribution (dN/dlogD) by a scanning mobility particle sizer, SMPS, (TSI

Inc., DMA Model 3080, CPC Model 3772; operated at 2.0/0.2 L min$^{-1}$ sheath/aerosol flow rates; 3-

130        min resolution) and an optical particle counter, OPC, (Grimm Inc., model 1.109, 655 nm operating

wavelength; flowrate 1.2 L min$^{-1}$, 6-s resolution). The SMPS measures the aerosol number

concentration in the electrical mobility diameter ($D_m$) range 0.019–0.882 µm, and the OPC

measures in the optical equivalent diameter ($D_{opt}$) range 0.25-32 µm.

Instrumental details are summarized in **Table 1**.
Sampling lines from the manifold to the instruments were made of conductive silicone tubing (TSI Inc.,
6.4·10$^{-3}$ m diameter) to minimize particle loss by electrostatic deposition. They were designed to be as
straight and as short as possible. Their length, varying between 0.3 and 0.7 m, was adjusted based
on the flowrate of each instrument to ensure an equivalent particle loss, so the same aerosol size
distribution could be assumed as input for all instruments. Particular care was given to ensure the
same aerosol size at the input of the aethalometer and the MAAP. To this end, as illustrated in Fig. 1,
the two instruments sampled air from the same manifold exit line, and also the same sampling flow
rate was set for the two instruments (8 L min$^{-1}$). Particle loss calculations were performed with the
Particle Loss Calculator (PLC) software (von der Weiden et al., 2009).
Aerosols were generated in three ways:
- mineral dust was generated by mechanical shaking as described and validated in Di Biagio et al.
(2014, 2017). About 3 gr of soil sample (sieved at 1000 µm and dried at 100°C) was placed in a
Büchner flask and shaken at 100 Hz by a sieve shaker (Retsch AS200). The dust was injected in the
manifold by a flow of N$_2$ at 3.5 L min$^{-1}$ through a single-stage impactor used to eliminate particles
larger than about 20 µm, which could be preferentially sampled by the instruments with the highest





flow rate. Pure $N_2$ was added to the aerosol flow to make the injection flow equal to the total sampling flow by instruments connected to the manifold (about 38 L min$^{-1}$);

- ammonium sulfate (Sigma-Aldrich 99.999% purity, 0.03 M solution in ultrapure water) and kaolinite particles (Source Clay Repository KGa-2, 0.05 M solution in ultrapure water) were generated by a constant flow atomizer (TSI, model 3075) operated at 3 L min$^{-1}$ and coupled with a diffusion drier (TSI, model 3062). As for dust, pure $N_2$ was added to the aerosol flow to equalize the total sampling flow;

- ambient pollution aerosols were sampled by opening the manifold to the exterior ambient air. Sampling was performed at the University Paris-Est Creteil, in the suburbs of Paris, at the ground floor of the University building, which is close to a main local road (~20 m) and to the A86 highway (~200 m).

**3. Strategy for data analysis**

The aethalometer spectral attenuation coefficient $\beta_{ATT}(\lambda)$ is related to the measured attenuation ATT($\lambda$) through the following formula:

$$\beta_{ATT}(\lambda) = \frac{\Delta ATT(\lambda)}{\Delta t} \frac{A}{V} \tag{1}$$

where A is the area of the aerosol collection spot (0.5 ± 0.1) cm$^2$ and V the air sampled volume (0.016 m$^3$ over 2-min integration time). $\Delta ATT(\lambda)/\Delta t$ in Eq. (1) can be calculated as the linear fit of the measured attenuation as a function of time.

The spectral attenuation coefficient $\beta_{ATT}(\lambda)$ measured by the aethalometer is related to the targeted absorption coefficient $\beta_{abs}(\lambda)$ by the following formula (C2010):

$$\beta_{abs}(\lambda) = \frac{\beta_{ATT}(\lambda) - \alpha(\lambda)\beta_{sca}(\lambda)}{R \cdot C_{ref}} \tag{2}$$

where the different terms parametrise different instrument artefacts:

- the scattering effect $\alpha(\lambda)\beta_{sca}(\lambda)$, that is, the amount of scattered radiation by the aerosols deposited on the filter that is miscounted as absorption, where $\alpha(\lambda)$ is a wavelength-dependent proportionality constant and $\beta_{sca}(\lambda)$ is the aerosol spectral scattering coefficient;

- the shadowing effect R, representing the artificial flattening of measured attenuation with time due to the gradual accumulation of absorbing particles on the loaded filter;

- the multiple scattering $C_{ref}$, representing multiple scattering of the light beam by the filter fibres.

The $\alpha(\lambda)$ term and R in Eq. (2) can be calculated through various empirical formulas reported in the literature (W2003, Arnott et al., 2005; Virkkula et al., 2007; Schmid et al., 2006; C2010). The determination of $C_{ref}$, instead, is the objective of our study.





### 3.1. Scattering effect correction

Arnott et al. (2005) provide for $\alpha(\lambda)$ the following formulation:

$$\alpha(\lambda) = A^{d-1} \cdot c \cdot \lambda^{-\alpha_s(d-1)} \tag{3}$$

where the A and $\alpha_S$ terms are obtained from the power-law fit of $\beta_{sca}(\lambda)$ versus $\lambda$, and the c and d terms can be determined from the power-law fit of the attenuation $\beta_{ATT}(\lambda)$ versus the scattering $\beta_{sca}(\lambda)$ coefficient as

$$\beta_{sca}(\lambda) = A\lambda^{-\alpha_s} \tag{4}$$

$$\beta_{ATT}(\lambda) = c\beta_{sca}(\lambda)^d \tag{5}$$

### 3.2. Shadowing effect correction

Two formulations for the shadowing effect correction R are proposed by W2003 and C2010:

$$R(W2003)(\lambda) = \left(\frac{1}{f(\lambda)} - 1\right)\frac{\ln(ATN(\lambda)) - \ln(10\%)}{\ln(50\%) - \ln(10\%)} + 1 \tag{6a}$$

$$R(C2010)(\lambda) = \left(\frac{1}{f(\lambda)} - 1\right)\frac{ATN(\lambda)}{50\%} + 1 \tag{6b}$$

The factor $f(\lambda)$ represents the dependence of the shadowing effect on the aerosol absorption. This dependence is parametrized by the aerosol single scattering albedo SSA($\lambda$) in the form of

$$f(\lambda) = a\left(1 - SSA(\lambda)\right) + 1 \tag{7}$$

where $a$, equal to 0.85 in W2003 and 0.74 in C2010, is obtained as the slope of the linear fit between the attenuation coefficient $\beta_{ATT}$ normalized to its value at 10% attenuation ($\beta_{ATT}/\beta_{10\%}$) and the natural logarithm of the measured attenuation $\ln(ATT(\lambda))$.

### 3.3. Multiple scattering correction

For the determination of $C_{ref}$ only $\beta_{ATT}$ and R are required. Henceforth in this work, attenuation data from the aethalometer were corrected for the shadowing effect R but not for the scattering term $\alpha(\lambda)\beta_{sca}(\lambda)$. Three different formulations of $C_{ref}$ were therefore considered:

$$C_{ref}^*(\lambda) = \frac{\beta_{ATT}(\lambda)}{\beta_{abs\text{-}ref}(\lambda)} \tag{8a}$$

$$C_{ref}(W2003)(\lambda) = \frac{1}{\beta_{abs\text{-}ref}(\lambda)}\frac{\beta_{ATT}(\lambda)}{R(W2003)(\lambda)} \tag{8b}$$

$$C_{ref}(C2010)(\lambda) = \frac{1}{\beta_{abs\text{-}ref}(\lambda)}\frac{\beta_{ATT}(\lambda)}{R(C2010)(\lambda)} \tag{8c}$$


The $\beta_{abs-ref}$ term in Eq. 8a-8c represents the reference absorption coefficient estimated from
independent measurements. $C_{ref}^{*}$ does not take into account the shadowing effect correction in
aethalometer data, as done by Schmid et al. (2006). $C_{ref}$(W2003) and $C_{ref}$(C2010) take this correction
into account, by using the R(W2003) and the R(C2010) parametrisations, respectively. The spectral
$\beta_{ATT}$/R(C2010) was used to calculate the absorption Ångström exponent ($\alpha_{A}$). Note that in this work
we considered, for each experiment, only data corresponding to ATT < 20% to calculate $\beta_{ATT}$
($R^2$>0.99 for the ΔATT/Δt fits in all cases, see Eq. (1)). This threshold was fixed based on two
requirements: first, we limited our data analysis to points with low attenuation in order to account
almost exclusively for the scattering by the filter fibers in the $C_{ref}$ calculation and not for the scattering
from aerosol particles embedded in the filter. This choice was done also for consistency with the
literature, since both W2003 and C2010 relate $C_{ref}$ to ATT~10%. Second, this choice ensured that
enough data points were available for analysis regardless of the aerosol type, in particular for ambient
aerosols, for which attenuation rapidly exceeded 10%.

### 220  3.4. Determination of reference absorption coefficient and single scattering albedo

The reference absorption coefficient $\beta_{abs-ref}$ in Eq. 8a-8c was obtained in different ways depending on
wavelength. At 450 nm, $\beta_{abs-ref}$ was obtained with the "extinction minus scattering" approach by using
the CAPS measurements for extinction and the nephelometer measurements for scattering. At 660
nm, $\beta_{abs-ref}$ was extrapolated from MAAP measurements at 670 nm.

### 225  3.4.1. Direct determination of reference absorption coefficient at 660 nm from the MAAP

The reference absorption coefficient $\beta_{abs-ref}$ at 660 nm was obtained by the MAAP measurement at
670 nm. The MAAP attenuation (ATT) at 670 nm is estimated from the measured transmission (T)
and retrieved single scattering albedo of the aerosol-filter layer ($SSA_0$, from the inversion algorithm)
as

$$ATT(670) = (1 - SSA_0) \cdot \ln T \cdot 100 \qquad (9)$$

Equation (1) is applied to estimate the absorption coefficient at 670 nm from ATT(670). The area of
the aerosol collection spot is 2 cm$^2$ and the sampled volume is 0.008 m$^3$ over 1-min integration time.
The absorption coefficient of the MAAP was extrapolated to the 660 nm wavelength by using the
absorption Ångström exponent $\alpha_A$ calculated from aethalometer data.

### 235  3.4.2. Indirect determination of reference absorption coefficient at 450 nm: "extinction minus
### 236  scattering" approach

The reference absorption coefficient $\beta_{abs-ref}$ at 450 nm was calculated as the difference between the
extinction and scattering coefficient from the CAPS and the nephelometer.
The extinction coefficient $\beta_{ext}$ at 450 and 630 nm was measured directly by the two CAPS analyzers
without additional corrections (Massoli et al., 2010). The spectral $\beta_{ext}$ was used to calculate the
extinction Ångström exponent ($\alpha_E$), applied then to extrapolate $\beta_{ext}$ at 660 nm.
The scattering coefficient $\beta_{sca}$ at 450, 550, and 700 nm measured by the nephelometer between 7 and
170° was corrected for the size-dependent angular truncation of the sensing volume to report it to the
full angular range 0°-180° (Anderson and Ogren, 1998). Two different approaches were used: for sub-
micrometric ammonium sulfate, the correction proposed by Anderson and Ogren (1998) was applied,
while for aerosols with a significant coarse fraction (dust, ambient air and kaolinite), the truncation
correction was estimated by optical calculations according to the Mie theory for homogeneous
spherical particles using as input the measured number size distribution. In the calculations the real
and the imaginary parts of the complex refractive index m (m=n-$i$k, where n is the real part and k is
the imaginary part) were varied in the wide range 1.42–1.56 and 0.001–0.025$i$ for dust (Di Biagio et
al., 2017), and 1.50–1.72 and 0.001–0.1$i$ for ambient air (Di Biagio et al., 2016), while the value of
1.56-0.001$i$ was assumed for kaolinite (Egan and Hilgeman, 1979; Utry et al., 2015). Then, n and k
were set to the values which reproduced the measured $\beta_{sca}$ at 7-170°. The truncation correction factor
($C_{trunc}$) was estimated as the ratio of the modelled $\beta_{sca}$ at 0°-180° and 7°-170°. At the three
nephelometer wavelengths (450, 550, and 700 nm) the correction factor $C_{trunc}$ varied in the range
1.03-1.06 for ammonium sulfate, 1.08-1.6 for dust, 1.03-1.05 for kaolinite, and 1.05-1.25 for ambient
air. Once corrected for truncation, the spectral $\beta_{sca}$ was used to calculate the scattering Ångström
exponent ($\alpha_S$), applied then to extrapolate $\beta_{sca}$ at 630 and 660 nm.
**3.4.3. Determination of the single scattering albedo (SSA)**
The aerosol single scattering albedo (SSA) represents the ratio of scattering to extinction. At 450 nm,
the SSA was estimated by nephelometer and CAPS data (Eq. 10), while at 660 nm CAPS data were
combined with MAAP observations (Eq. 11):
$$SSA(450)=\frac{\beta_{sca}(450)_{nephelometer}}{\beta_{ext}(450)_{CAPS}} \tag{10}$$

$$SSA(660)=\frac{\beta_{ext}(660)_{CAPS}-\beta_{abs-MAAP}(660)}{\beta_{ext}(660)_{CAPS}} \tag{11}$$


**3.5. Number size distribution and effective fine and coarse diameter**
The number size distribution was measured by a combination of SMPS and OPC observations. For
the SMPS, corrections for particle loss by diffusion in the instrument tubing and the contribution of
multiple-charged particles were performed using the SMPS software. The electrical mobility diameter
measured by the SMPS can be converted to a geometrical diameter ($D_g$) by taking into account the
particle dynamic shape factor ($\chi$; $D_g=D_m/\chi$). In this study, the SMPS showed a good agreement with
OPC data for a shape factor $\chi=1$, which corresponds to spherical particles.
The OPC optical-equivalent nominal diameters were converted into sphere-equivalent geometrical
diameters ($D_g$) by taking into account the aerosol complex refractive index. For dust aerosols the
refractive index was varied in the range 1.47-1.53 (n) and 0.001-0.005$i$ (k) and $D_g$ was set at the
mean ± one standard deviation of the values obtained for the different n and k. For kaolinite the OPC
diameter conversion was performed by setting the refractive index at 1.56-0.001$i$. For ambient air the



refractive index was set at 1.60-0.01$i$, a value that represents a medium absorbing urban polluted
aerosol (see Di Biagio et al., 2016). After conversion, the OPC diameter range became 0.28-18.0 μm
for dust (taking into account the particle cut at ~20 μm due to the use of the impactor), and 0.27-58.0
μm for kaolinite and 0.28-65.1 μm for ambient air (the impactor was not used in these cases). The
uncertainty was <15% at all diameters.
The aerosol effective fine ($D_{eff,fine}$) and coarse ($D_{eff,coarse}$) diameter were estimated from OPC data as

$$D_{eff,fine} = \frac{\int_{0.3\mu m}^{1.0\mu m} D_g^3 \frac{dN}{dlogD_g} dlogD_g}{\int_{0.3\mu m}^{1.0\mu m} D_g^2 \frac{dN}{dlogD_g} dlogD_g}$$

(12)


$$D_{eff,coarse} = \frac{\int_{1\mu m}^{10\mu m} D_g^3 \frac{dN}{dlogD_g} dlogD_g}{\int_{1\mu m}^{10\mu m} D_g^2 \frac{dN}{dlogD_g} dlogD_g}$$

13)

### 3.6. Data integration and error analysis

Aethalometer data were first processed at 2-min resolution to obtain the time evolution of the
attenuation coefficients $\beta_{ATT}$ and $\beta_{ATT}/R$. Data from the MAAP, CAPS, nephelometer, OPC and SMPS
were averaged over 2-min to report them to the same resolution of the aethalometer.
Then the $\beta_{ATT}$ and $\beta_{ATT}/R$ were calculated over the whole duration of each experiment from Eq. (1)
and (6). Corresponding averages of the reference absorption coefficient ($\beta_{abs-ref}$) were calculated for
each experiment and used to estimate $C_{ref}$. Experiment-averages of SSA, $D_{eff,fine}$, and $D_{eff,coarse}$ were
also calculated to relate to the obtained $C_{ref}$.
The uncertainty on $C_{ref}$ was estimated with the error propagation formula by taking into account the
uncertainties on $\beta_{ATT}$, $\beta_{ATT}/R$, and the standard deviation of the averaged $\beta_{abs-ref}$ from the CAPS-
nephelometer and the MAAP. The uncertainty on $\beta_{ATT}$ was estimated as the quadratic combination of
the uncertainty on the linear fit of ΔATT with respect to time and the uncertainties on the surface
deposit A. The uncertainty on $\beta_{ATT}/R$ was estimated taking into account the uncertainty on $\beta_{ATT}$ and R.
Uncertainties on $\beta_{ATT}$ and $\beta_{ATT}/R$ are both ~20%.

### 4. Results

The time series of observations for all the experiments are shown in **Fig. 2** as 2-min averages. Seven
experiments were performed on mineral dust issued from six different areas in the Sahel (Niger),
Eastern Asia (China), North America (Arizona), Northern Africa (Tunisia), Australia, and Southern
Africa (Namibia), and on a kaolinite powder. Experiments were performed between the 3[rd] and the 9[th]
of November 2016 and lasted between 1 and 2 hours each. The experiment on Niger dust (labelled
as Niger 1 and Niger 2) were duplicated to test the repeatability of the obtained $C_{ref}$. Ambient air data
were collected between the 8[th] and the 14[th] November 2016 for a total of 7 hours of measurements.
Eight different periods characterized by little variation and different levels of SSA were selected in the





whole set of ambient air measurements. These are identified as ambient air 1 to 8. The summary of
information is provided in **Table 2**. SMPS data were available for ammonium sulfate and kaolinite
experiments, for one of the two Niger dust experiments (Niger 2), and for some of the ambient air
experiments. OPC measurements were performed for all experiments with the exception of the
ammonium sulfate.

### 4.1. Quality control data

Results of the ammonium sulfate control experiment (24 October 2016), used to test the
performances of the optical instruments, are illustrated in **Fig. 3**. As expected for this purely scattering
aerosol (Toon et al., 1976), the nephelometer scattering and the CAPS extinction at 450 and 630 nm
were in very good agreement (less than 4% difference) during the whole duration of the experiment.
This is further explicated by the scatterplot of their respective 10-minute averages, yielding a linear
regression in the form of y=0.95x+5.1 ($R^2$=0.95) at 450 nm and y=1.01x-1.4 ($R^2$=0.98) at 630 nm. The
average $\beta_{ext}$ at 450 and 630 nm from CAPS observations was 913 (± 52) and 424 (± 33) Mm$^{-1}$,
respectively, while the average $\beta_{sca}$ was 921 (± 36) and 420 (± 17). This led to an average SSA of
1.01 (± 0.07) at 450 nm and 0.99 (± 0.07) at 630 nm.
The absorption coefficient, averaged over the duration of the experiment, was 0.10 (± 0.04) Mm$^{-1}$ at
450 nm and 0.24 (± 0.07) Mm$^{-1}$ at 660 nm according to the aethalometer, and 0.82 (± 0.13) Mm$^{-1}$ at
660 nm according to the MAAP. For the aethalometer, the absorption coefficient was calculated from
Eq. (2) assuming $C_{ref}$=2.14 and the R formulation by C2010 (Eq. 6b). The $\alpha(\lambda)$ coefficient was
calculated from Eq. (3). The c and d terms in Eq. (3) were determined from the power-law fit of $\beta_{ATT}(\lambda)$
vs $\beta_{sca}(\lambda)$ and are c=(0.56 ± 0.06) Mm$^{-1}$ and d=(0.485 ± 0.09). These values are lower than those
reported by Arnott et al. (2005) (c=0.797, d=0.564). The A and $\alpha_S$ terms, obtained from the power law
fit of $\beta_{sca}(\lambda)$ vs wavelength (Eq. 3) are A=(4.07 ± 0.49)10$^9$ Mm$^{-1}$ and $\alpha_S$=(-2.46 ± 0.12).
**Figure 4** shows the extinction coefficient at 660 nm extrapolated from CAPS observations and
calculated as the sum of nephelometer and MAAP data for dust, kaolinite, and ambient air
experiments. The linear regression of the data yields y=1.03x-0.5 ($R^2$=0.99), indicating the
consistency of optical measurements between the CAPS, nephelometer, and MAAP (less than 3%
difference on average). Based on the success of the optical closure at 660 nm, we therefore assume
the "CAPS minus nephelometer" approach appropriate to estimate the aerosol absorption coefficient
at 450 nm.

### 4.2. Estimate of $C_{ref}$

The $C_{ref}^*$, $C_{ref}$(W2003) and $C_{ref}$(C2010) at 450 and 660 nm obtained for all different experiments and
the corresponding aerosol SSA, $D_{eff,fine}$, and $D_{eff,coarse}$ are summarized in **Table 2**.
$C_{ref}$ for mineral dust varied between 1.81 and 2.56 for a SSA of 0.85–0.96 at 450 nm and between
1.75 and 2.28 for a SSA of 0.98–0.99 at 660 nm. The estimate for Niger 1 and 2 samples agreed
within 4.9%, which suggests a good repeatability of the $C_{ref}$ estimate. For kaolinite $C_{ref}$ was 2.47–2.51
and 2.31–2.34 at 450 and 660 nm, respectively, with an associated SSA of 0.96 and 0.97 at the two
wavelengths. For ambient air $C_{ref}$ varied in the range 1.91–4.35 for a SSA of 0.62–0.87 at 450 nm and



1.66–2.96 for and SSA of 0.42–0.76 at 660 nm. For samples 6 and 8 the $C_{ref}$ at 450 was lower than at
660 nm. Otherwise, for all other cases, the $C_{ref}$ was larger at 450 nm than at 660 nm.
Differences within 2.8% were obtained between $C_{ref}^*$, $C_{ref}$(W2003) and $C_{ref}$(C2010) at 450 and 660 nm
for weakly-absorbing dust and kaolinite. Instead, for more absorbing ambient air aerosols the
differences between $C_{ref}^*$, $C_{ref}$(W2003) and $C_{ref}$(C2010) were in the range 2.7% to 24.3%. In some
cases (ambient air 1–2 and Niger 1 samples) we obtained $C_{ref}$(C2010)>$C_{ref}$(W2003); these cases
correspond to a mean aethalometer measured ATT<10%, for which R(W2003)>R(C2010), and this
explains the larger $C_{ref}$(C2010). Conversely, $C_{ref}$(C2010)<$C_{ref}$(W2003) when the measured ATT was
~15-20%, yielding R(W2003)<R(C2010). The percent difference between the obtained $C_{ref}$(W2003)
and $C_{ref}$(C2010) increased for decreasing SSA due to the increase of the R(W2003) to R(C2010)
absolute difference for decreasing SSA. When averaging data for all ambient air samples, the two
formulations yield very similar values. For example, at 660 nm the mean $C_{ref}$(W2003) was 2.44 (±
0.38), less than 2% larger than the mean $C_{ref}$(C2010) of 2.39 (± 0.35).
The different ATT threshold assumed here (20%) compared to W2003 and C2010 (10%) has a
negligible impact (less than 1% difference) on the results.
The mean and standard deviation of the multiple scattering correction at 450 and 660 nm for dust,
kaolinite, and ambient air calculated as the mean of the $C_{ref}^*$, $C_{ref}$(W2003), and $C_{ref}$(C2010) is reported
in **Table 3**. The mean $C_{ref}$ at 450 and 660 nm is 2.09 (± 0.22) and 1.92 (± 0.17) for dust, 2.49 (± 0.02)
and 2.32 (± 0.01) for kaolinite, and 2.32 (± 0.36) and 2.32 (± 0.35) for pollution aerosols
**4.3. Dependence of $C_{ref}$ on SSA**
As reported in Table 2, very different SSA values at 450 and 660 nm were obtained for the various
cases. For dust aerosols, the measured SSA values were larger than 0.85 at 450 nm and close to
unity (>0.98) at 660 nm, in line with field observations of dust from different sources (Schladitz et al.,
2009; Formenti et al., 2011; Ryder et al., 2013). The SSA for kaolinite was 0.96–0.97 at 450 and 660
nm, in agreement with Utry et al. (2016) (0.97 and 0.99 (±0.04) at 450 and 635 nm, respectively). Both
at 450 and 660 nm, the single scattering albedo for ambient air varied in the wide range 0.2 to 0.9
during the whole measurement period (see Fig. 2 for measurements at 660 nm). The average values
obtained for air samples 1–8 were 0.62–0.87 at 450 and 0.42–0.76 at 660 nm. The SSA decreased
with increasing wavelength, as expected for pollution aerosols (e.g., Bergstrom et al., 2007; Di Biagio
et al., 2016). The wide range of values indicates the occurrence of particles with very different
absorption properties, henceforth chemical composition. For instance, in urban environments,
Bergstrom et al. (2007) reported SSA in the range 0.2–1.0 at 550 nm, with lowest values observed for
soot-dominated air masses and highest values for urban pollution dominated by low-absorbing
organic components.
The experimental SSA values served to two purposes. First, as shown in **Fig. 5**, they are linearly
related to the factor f in the shadowing effect correction term R in Eq. (6a)-(6b) as f=$a$(1-SSA)+1. The
linear regression of our data yields a slope $a$=(1.48 ± 0.14), larger than the value of 0.85 reported in
W2003 (f data from W2003 are also shown in Fig. 5) and 0.76 in C2010.





Secondly, SSA data serve to investigate the dependence of $C_{ref}$ on particle absorption for mineral
dust. As shown in **Fig. 6** (top panel), $C_{ref}$ for dust seems to be independent of SSA at 660 nm,
whereas it decreases for increasing SSA at 450 nm. This trend is statistically significant (correlation
coefficient of $R^2$=0.85). The relationship between $C_{ref}$ and SSA is also investigated in **Fig. 6** (bottom
panel) for all aerosol samples. Globally, Fig. 6 suggests a decrease of $C_{ref}$ for increasing SSA, in
particular at 450 nm, albeit with a poorer statistical significance at both wavelengths ($R^2$=0.35 and
0.59). Data are also compared to those reported in W2003 and C2010 at 660 nm for different aerosol
types. Diesel soot and soot mixed with ammonium sulfate were investigated in W2003, while C2010
reported data for ambient aerosols sampled at different locations in Europe and in Amazonia. W2003
also reported the $C_{ref}$ for soot particles at 450 nm (not shown in Fig. 6), with values between 2.08 and
3.64; these values are in line with our observations at 450 nm for ambient air. Whereas, as illustrated
in Fig. 6, both W2003 and C2010 found a relationship between $C_{ref}$ and SSA at 660 nm, contrasting
results are obtained when plotting the two datasets together. C2010 obtained a sharp and almost
linear decrease of $C_{ref}$ with increasing SSA ($C_{ref}$~5–2.5 for SSA~0.65–0.9), while W2003 data showed
a pronounced decrease of $C_{ref}$ (~2–4) for increasing SSA in the range 0.5 and 0.7 and low $C_{ref}$ values
(~2) at SSA~0.2. Our data for dust and kaolinite at high SSA (>0.97) seem to follow the same linear
relationship as C2010. However at lower SSA, our data for ambient aerosols are closer to W2003
results at 660 nm. These differences between W2003 and C2010 data, and also with our results, are
quite difficult to explain. The main difference between W2003 compared to C2010 is that W2003
performed measurements in a simulation chamber, while C2010 was a field study. Working in
ambient conditions may influence the retrieved $C_{ref}$. In fact, volatile-organic compounds or water vapor
present in the atmosphere may condense on the filter, thus enhancing the scattering from the filter
fibers and leading to higher $C_{ref}$. This could explain the higher $C_{ref}$ obtained in C2010 compared to
W2003. Our results for ambient air particles, however, are in agreement with W2003 chamber results.
Differences in the size distributions of the investigated aerosols are also expected to possibly affect
the comparison; however, no detailed information on the size of investigated aerosols is provided in
W2003 and C2010. Another source of discrepancy may be in the fact that, differently from W2003 and
our study, where aethalometer and MAAP were compared at 660 nm, $C_{ref}$ in C2010 was estimated by
comparing aethalometer data at 660 nm with MAAP observations at 630 nm. As aerosol absorption
increases with decreasing wavelength, this wavelength difference may induce an underestimation of
$C_{ref}$ in C2010.
**4.4. Dependence of $C_{ref}$ on particles size**
Examples of the number size distribution measured by the SMPS and OPC for ammonium sulfate,
Niger dust, kaolinite, and ambient air are shown in **Fig. 7**. Ammonium sulfate had mostly a submicron
distribution, while dust aerosols presented the largest fraction over the whole super-micron range up
to about 10-20 µm. Dust particles larger than 20 µm were completely suppressed by the impactor
system and were not detected by the OPC. The coarse component, up to about 10 µm, was also
identified in the kaolinite and ambient air samples. In particular, a defined mode at ~4 µm was
detected in ambient air particles, may be linked to the presence of soot-aggregates, tire abrasions, re-
suspended road dust, or bioaerosols (Harrison et al., 2001; Bauer et al., 2008; Pakbin et al., 2010; Liu



and Harrison, 2011). In correspondence, the $D_{eff,fine}$ varied between 0.24 and 0.62 µm and the $D_{eff,coarse}$ between 2.3 and 6.2 µm for the different cases (Table 2). For mineral dust, $D_{eff,coarse}$ ranged between 2.3 and 3.6 µm, encompassing the value of $D_{eff,coarse} \sim 3$ µm reported by Denjean et al. (2016b) in their figure 11 for Saharan dust both close to sources and during transport over the Atlantic.

These observations are consistent with the extinction ($\alpha_E$) and the absorption ($\alpha_A$) Ångstrom exponent measured during the experiments. The $\alpha_E$ (shown in Fig.3) was ~1 for kaolinite, varied between about 1 and 2 for mineral dust aerosols, and between 0.5 and 2.5 for ambient air, indicating particles with variable sizes, both the sub-micron and the super-micron fractions. The absorption Ångström coefficient $\alpha_A$ obtained from aethalometer data was between 2.2 and 3 for dust, and about 1 for kaolinite and ambient air aerosols.

The dependence of $C_{ref}$ at 450 and 660 nm on the effective diameter fine $D_{eff,fine}$ and coarse $D_{eff,coarse}$ as a measure of particle size was investigated. The scatterplot of $C_{ref}$ versus $D_{eff,coarse}$ is shown in **Fig. 8** and indicates that the $C_{ref}$ does not have any statistically significant dependence on the particle size for mineral dust at both wavelengths and for all data at 660 nm ($R^2 \leq 0.40$). Conversely, a slight increase of $C_{ref}$ for increasing $D_{eff,coarse}$ is obtained at 450 nm when all aerosol samples are considered ($R^2 = 0.70$). No dependence of $C_{ref}$ versus $D_{eff,fine}$ is instead obtained for all cases ($R^2 \leq 0.44$) (not shown).

**5. Conclusions**

In this paper we presented an intercomparison study between an aethalometer and a MAAP, a nephelometer, and two CAPS with the aim of determining a two-wavelength multiple scattering correction ($C_{ref}$) for aethalometer measurements for weakly-absorbing mineral dust aerosols. Mineral dust aerosols investigated here were generated from natural parent soils collected in desert areas, both in the Northern and in the Southern hemisphere (Di Biagio et al., 2014; 2017). The size distribution of the generated dust included both the submicron and the supermicron fractions, with an effective fine and coarse diameter between 0.32–0.55 and 2.3–3.6 µm, respectively.

The estimated $C_{ref}$ was in the range 1.81–2.56 at 450 nm and 1.75–2.28 at 660 nm for the different dust samples, with mean $C_{ref}$ values of 2.09 (± 0.22) and 1.92 (± 0.17), respectively. The dust absorption coefficient estimated by the aethalometer should henceforth be about 2% (450 nm) and 11% (660 nm) higher than obtained by using the wavelength-independent value of 2.14, commonly used in the literature (e.g., Sandradewi et al., 2008; Formenti et al., 2011; Di Biagio 2016). The new estimate of $C_{ref}$ has a negligible impact on the dust SSA at 450 nm (less than 0.5% difference between the value obtained for $C_{ref}$=2.09 or 2.14), but affects by up to ~3% the estimate of SSA at 660 nm.

Given that the maximum intensity of the solar spectrum occurs at about 700 nm, the expected change in the dust SSA at 660 nm may significantly affect the impact of dust on radiation. Mallet et al. (2009) estimated that about a 3% change in the visible SSA of dust may determine up to a 10% change in





the radiative effect of dust at the surface, and up to 20% change at the Top of the Atmosphere, with a
net ~25% increase of dust absorption in the atmosphere. Given the strong sensitivity of the dust direct
effect to particle absorption (Solmon et al., 2008; Mallet et al., 2009; Di Biagio et al., 2010; Jin et al.,
2016, among others), we recommend this new $C_{ref}$ value at 660 nm to be used when analyzing
aethalometer data for mineral dust aerosols.
The analysis performed in this study indicates that there is no dependence of $C_{ref}$ on the coarse
component of the particle size distribution for dust. This suggests that the $C_{ref}$ obtained here can be
used to correct aethalometer data for dust at emission, when the coarse fraction dominates the dust
size distribution, as well as after long-range transport, when the coarsest component of dust has
preferentially settled out.
Finally, even if beyond the scope of the paper, our body of observations, spanning a wide range of
SSA values from 0.96–0.97 (kaolinite) to ~0.4–0.8 (ambient urban aerosols), indicates that $C_{ref}$
decreases for increasing SSA, both at 450 and 660 nm. This is generally consistent with the results of
W2003 and C2010 at 660 nm. However, a unique relationship cannot be established. At high SSA
(>0.90), our data, as well as those of C2010, suggest a sharper decrease than at SSA in the range
0.4-0.8, where our data are more consistent with those of W2003. Differences in aerosol sampling
conditions and in the exact analysed wavelengths from the three studies may be the cause of such
discrepancy, but clear conclusions, as well as an explicit relationship between $C_{ref}$ and SSA, are still
difficult to give. Similarly, our observations seem to indicate that $C_{ref}$ increases for increasing $D_{eff,coarse}$
at 450 nm. This trend was however observed only when the whole aerosol dataset was considered,
and not if we limited to dust observations, so making difficult to draw clear conclusions.
A more extensive characterization of $C_{ref}$ should be required to provide an appropriate correction of
aethalometer data under the wide range of atmospheric conditions.

**Author contributions**

C. Di Biagio and P. Formenti designed the experiments, discussed the results, and wrote the
manuscript with comments from all co-authors. N. Marchand provided the MAAP used in the
experiments. C. Di Biagio, M. Cazaunau, and E. Pangui performed the experiments. C. Di Biagio
performed the data analysis.

**Acknowledgements**

The RED-DUST project was supported by the French national programme LEFE/INSU, by the Institut
Pierre Simon Laplace (IPSL), and by OSU-EFLUVE (Observatoire des Sciences de l'Univers-
Enveloppes Fluides de la Ville à l'Exobiologie) through dedicated research funding. C. Di Biagio was
supported by the CNRS via the Labex L-IPSL, which is funded by the ANR (grant no. ANR-10-LABX-
0018). This work has also received funding from the European Union's Horizon 2020 (H2020)
research and innovation programme through the EUROCHAMP-2020 Infrastructure Activity under



grant agreement No 730997. The authors thank K. Kandler, D. Seibert, and the LISA staff who
collected the soil samples used in this study, E. Journet who provided the kaolinite sample, A. Petzold
for helpful discussions on the aethalometer multiple scattering effects, and B. Tamime-Roussel for
logistic help with the MAAP.

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





**Table captions**

**Table 1.** Specifications and references of instruments used during experiments.

**Table 2**. Summary of experiments and results. The mean and the standard deviation of $D_{eff,fine}$, $D_{eff,coarse}$, SSA at 450 and 660 nm, $C_{ref}^{*}$, $C_{ref}(W2003)$, and $C_{ref}(C2010)$ are reported. As a reminder: $C_{ref}^{*}$ is the multiple scattering correction obtained not taking into account the shadowing effect correction in aethalometer data; $C_{ref}(W2003)$ and $C_{ref}(C2010)$ take the shadowing effect correction into account, by using the parametrisations by Weingartner et al. (2003) (referred as W2003) and Collaud Coen et al. (2010) (referred as C2010), respectively. The maximum of the % difference between $C_{ref}^{*}$, $C_{ref}(W2003)$, and $C_{ref}(C2010)$ is indicated in the table.

**Table 3**. Mean and standard deviation multiple scattering correction $\overline{C_{ref}}$ at 450 and 660 nm for dust, kaolinite, and ambient air. The $\overline{C_{ref}}$ was calculated as the mean of the $C_{ref}^{*}$, $C_{ref}(W2003)$, and $C_{ref}(C2010)$ obtained at each wavelength for the different aerosol types. As a reminder: $C_{ref}^{*}$ is the multiple scattering correction obtained not taking into account the shadowing effect correction in aethalometer data; $C_{ref}(W2003)$ and $C_{ref}(C2010)$ take the shadowing effect correction into account, by using the parametrisations by Weingartner et al. (2003) and Collaud Coen et al. (2010), respectively.

**Figure captions**

**Figure 1.** Experimental setup used for the aethalometer intercomparison experiments.

**Figure 2.** Temporal series of experiments showing the measured optical data at 660 nm. The different panels show (from the top to the bottom): (i) the shadowing-corrected aethalometer attenuation at 660 nm (data corrected with the R formulation by Collaud Coen et al. (2010) (referred as R(C2010)) are shown) and the MAAP aerosol absorption coefficient; (ii) the aerosol extinction at 660 nm extrapolated from CAPS PMex measurements and estimated as the sum of nephelometer scattering and MAAP absorption; (iii) the extinction aerosol Ångstrom exponent; (iv) the aerosol single scattering albedo at 660 nm. Each point in the plot corresponds to 2 min average data. The x-axis indicates the data point sequential number. Experiments with dust samples and kaolinite were realised between the 3$^{rd}$ and the 9$^{th}$ of November 2016 and lasted between 1 and 2 hours each. Ambient air data were collected at different steps between the 8$^{th}$ and the 14$^{th}$ November 2016 for a total of 7 hours of measurements.

**Figure 3.** Ammonium sulfate experiment. Left panel: temporal evolution of the extinction and scattering coefficients measured by the CAPS PMex and the nephelometer at 450 nm (blue scale) and 630 nm (red scale). Each point in the plot corresponds to 2 min average data. Right panel: CAPS PMex versus nephelometer data (10 minutes averages). The y=x line and the results of the linear fit between CAPS and nephelometer data are also shown in the plot.

**Figure 4.** CAPS PMex extinction coefficient extrapolated at 660 nm versus nephelometer+MAAP calculated extinction at 660 nm for all experiments (dust, kaolinite, ambient air). Each point in the plot





corresponds to 10 min average data. The y=x line and the results of the linear fit between CAPS and
nephelometer+MAAP data are also shown in the plot.
**Figure 5.** Left panel: estimated f values versus (1-SSA) at 660 nm for dust aerosols. Different
symbols are used to distinguish between dust from different sources. Right panel: f versus SSA at
660 nm for all experiments. Different symbols are used to distinguish between different aerosol types.
The results of the linear fit between f and (1-SSA) are also reported. Data from Weingartner et al.
(2003) (W2003) (extracted from their Figure 4) are also shown in the plot for comparison.
**Figure 6.** Top panel: $C_{ref}$(W2003) (multiple scattering correction obtained by taking into account the
shadowing effect correction using the parametrisations by Weingartner et al. (2003))  versus SSA at
450 and 660 nm for mineral dust samples analysed in this study. Different symbols are used to
distinguish between dust from different sources. As indicated in Table 2, the difference between $C_{ref}^{*}$,
$C_{ref}$(W2003), and $C_{ref}$(C2010) is very low for mineral dust aerosols. Bottom panel: $C_{ref}$ versus SSA at
450 and 660 nm for the different aerosol samples analysed in this study. Different symbols are used
to distinguish between different aerosol types. Data for both $C_{ref}$(W2003) and $C_{ref}^{*}$ (multiple scattering
correction obtained not taking into account the shadowing effect correction in aethalometer data) are
shown for ambient air aerosols, while for dust and kaolinite, for which the difference between the
different formulations is very low, only $C_{ref}$(W2003) is reported. Data from Weigartner et al. (2003)
(W2003) ($C_{ref}$ from their Table 3, and SSA extracted from their Fig. 4) and Collaud Coen et al. (2010)
(C2010) (extracted from their Fig. 5) at 660 nm are also shown in the plot for comparison. The results
of the linear fits between $C_{ref}$ and SSA for mineral dust and for the entire dataset are also shown in the
plot.
**Figure 7.** Examples of number size distribution (normalised to the total number concentration) for
ammonium sulfate, dust (Niger sample), kaolinite, and ambient air aerosols. Data refer to the mean
over each experiment as measured from the SMPS and the OPC. Error bars (standard deviations)
have been omitted for the sake of clarity.
**Figure 8.** Top panel: $C_{ref}$(W2003) (multiple scattering correction obtained by taking into account the
shadowing effect correction using the parametrisations by Weingartner et al. (2003)) at 450 and 660
nm versus the effective diameter coarse $D_{eff,coarse}$. for mineral dust samples analysed in this study.
Different symbols are used to distinguish between dust from different sources. Bottom panel: $C_{ref}$ at
450 and 660 nm versus the effective diameter coarse $D_{eff,coarse}$ for the different aerosol samples
analysed in this study. Different symbols are used to distinguish between different aerosol types. Data
for both $C_{ref}$(W2003) and $C_{ref}^{*}$ (multiple scattering correction obtained not taking into account the
shadowing effect correction in aethalometer data) are shown for ambient air aerosols, while for dust
and kaolinite, for which the difference between the different formulations is very low, only $C_{ref}$(W2003)
is reported. The results of the linear fits between $C_{ref}$ and $D_{eff,coarse}$ for mineral dust and for the entire
dataset are also shown in the plot.





**Table 1.** Specifications and references of instruments used during experiments.

| Instrument | Property | Operating wavelength (nm) | Time resolution | Flowrate (L min$^{-1}$) | Percent uncertainty | Reference |
|---|---|---|---|---|---|---|
| Aethalometer (model AE-31, Magee Sci.) | Spectral absorption coefficient | 370, 470, 520, 590, 660, 880, 950 | 2 min | 8 | ±20% (attenuation coefficient) | Hansen et al. (1984) ; W2003 ; C2010 |
| Multi-Angle Absorption Photometer (MAAP, model 5012, Thermo Sci.) | Single-wavelength absorption coefficient | 670 | 1 min | 8 | ±12% | Petzold and Schönlinner (2004); Petzold et al. (2004 and 2005) |
| Cavity Attenuated Phase Shift Extinction (CAPS PMex, Aerodyne) | Spectral extinction coefficient | 450, 630 | 1 s | 0.85 | ±5% | Massoli et al. (2010) |
| Nephelometer (model 3563, TSI Inc.) | Spectral scattering coefficient | 450, 550, 700 | 1 s | 18 | ±10% | Anderson and Ogren (1998) |
| SMPS (DMA model 3080, CPC model 3772, TSI Inc.) | Number size distribution | – | 3 min | 2 | – | De Carlo et al. (2004) |
| OPC optical particle counter (model 1.109, Grimm Inc.) | Number size distribution | 655 | 6 s | 1.2 | ±15% (diameter optical to geometric conversion); ±10 (concentration) | Heim et al. (2008) |







**Table 2**. Summary of experiments and results. The mean and the standard deviation of $D_{eff,fine}$,
$D_{eff,coarse}$, SSA at 450 and 660 nm, $C_{ref}^*$, $C_{ref}$(W2003), and $C_{ref}$(C2010) are reported. As a reminder:
$C_{ref}^*$ is the multiple scattering correction obtained not taking into account the shadowing effect
correction in aethalometer data; $C_{ref}$(W2003) and $C_{ref}$(C2010) take the shadowing effect correction
into account, by using the parametrisations by Weingartner et al. (2003) (referred as W2003) and
Collaud Coen et al. (2010) (referred as C2010), respectively. The maximum of the % difference
between $C_{ref}^*$, $C_{ref}$(W2003), and $C_{ref}$(C2010) is indicated in the table.

| Aerosol ID | Source | $D_{eff,fine}$ (μm) $D_{eff,coarse}$ (μm) | SSA *450 nm* *660 nm* | $C_{ref}^*$ *450 nm* *660 nm* | $C_{ref}$ (W2003) *450 nm* *660 nm* | $C_{ref}$ (C2010) *450 nm* *660 nm* | Max % diff $C_{ref}$ *450 nm* *660 nm* |
|---|---|---|---|---|---|---|---|
| Ammonium sulfate | Sigma-Aldrich 99.999% purity | – | *0.999 ± (<)0.001* 0.999 ± (<)0.001 | | – | – | |
| Niger 1 | Sahel (13.52°N, 2.63°E) | 0.38 ± 0.01 2.6 ± 0.1 | *0.93 ± 0.01* 0.98 ± 0.01 | *2.00 ± 0.45* 1.87 ± 0.51 | *2.01 ± 0.45* 1.87 ± 0.51 | *2.02 ± 0.45* 1.88 ± 0.51 | *1.0 %* 0.4 % |
| Niger 2 | Sahel (13.52°N, 2.63°E) | 0.32 ± 0.02 2.3 ± 0.1 | *0.92 ± 0.01* 0.98 ± 0.01 | *2.05 ± 0.46* 1.89 ± 0.57 | *2.11 ± 0.47* 1.92 ± 0.56 | *2.10 ± 0.47* 1.92 ± 0.57 | *2.8 %* 1.6 % |
| China | Gobi desert (39.43°N, 105.67°E) | 0.44 ± 0.01 3.1 ± 0.2 | *0.94 ± 0.01* 0.98 ± 0.01 | *2.15 ± 0.48* 2.02 ± 0.62 | *2.16 ± 0.48* 2.01 ± 0.62 | *2.16 ± 0.48* 2.02 ± 0.63 | *0.5 %* 0.3 % |
| Arizona | Sonoran desert (33.15°N, 112.08°W) | 0.53 ± 0.02 3.1 ± 0.2 | *0.96 ± 0.01* 0.99 ± 0.01 | *1.81 ± 0.40* 1.76 ± 0.56 | *1.82 ± 0.41* 1.78 ± 0.55 | *1.82 ± 0.41* 1.78 ± 0.57 | *0.5 %* 1.1 % |
| Tunisia | Sahara desert (33.02°N, 10.67°E) | 0.48 ± 0.03 3.2 ± 0.7 | *0.96 ± 0.01* 0.99 ± 0.01 | *1.97 ± 0.49* 1.80 ± 0.42 | *1.98 ± 0.44* 1.80 ± 0.42 | *1.98 ± 0.44* 1.80 ± 0.42 | *0.5 %* 0 % |
| Australia | Strzelecki desert (31.33°S, 140.33°E) | 0.55 ± 0.02 2.4 ± 0.1 | *0.85 ± 0.01* 0.98 ± 0.01 | *2.52 ± 0.56* 2.28 ± 0.74 | *2.56 ± 0.57* 2.26 ± 0.72 | *2.56 ± 0.57* 2.28 ± 0.74 | *1.6 %* 0.9 % |
| Namibia | Namib desert (19.0°S, 13.0°E) | 0.45 ± 0.04 3.6 ± 0.3 | *0.95 ± 0.01* 0.98 ± 0.01 | *2.02 ± 0.45* 1.75 ± 0.57 | *2.03 ± 0.45* 1.76 ± 0.54 | *2.03 ± 0.45* 1.79 ± 0.57 | *0.5 %* 2.2 % |
| Kaolinite | Source Clay Repository KGa-2 | 0.39 ± 0.07 2.3 ± 1.6 | *0.96 ± 0.01* 0.97 ± 0.01 | *2.47 ± 0.55* 2.31 ± 0.60 | *2.51 ± 0.56* 2.34 ± 0.60 | *2.50 ± 0.56* 2.33 ± 0.60 | *1.6 %* 1.3 % |
| Ambient air 1 | Suburbs of Paris | 0.24 ± 0.08 5.2 ± 0.9 | *0.79 ± 0.05* 0.61 ± 0.08 | *3.87 ± 0.87* 1.97 ± 0.71 | *4.01 ± 0.90* 2.05 ± 0.73 | *4.03 ± 0.90* 2.11 ± 0.76 | *4.0 %* 6.6 % |
| Ambient air 2 | Suburbs of Paris | 0.50 ± 0.02 4.5 ± 0.1 | *0.72 ± 0.04* 0.67 ± 0.09 | *3.22 ± 0.72* 1.66 ± 0.44 | *3.68 ± 0.82* 1.94 ± 0.52 | *3.57 ± 0.80* 1.87 ± 0.50 | *12.5 %* 14.4 % |
| Ambient air 3 | Suburbs of Paris | 0.46 ± 0.03 6.2 ± 0.7 | *0.78 ± 0.06* 0.54 ± 0.10 | *3.93 ± 0.88* 2.32 ± 0.76 | *4.35 ± 0.97* 2.78 ± 0.89 | *4.25 ± 0.95* 2.68 ± 0.87 | *21.1 %* 16.5 % |
| Ambient air 4 | Suburbs of Paris | 0.53 ± 0.05 5.3 ± 1.3 | *0.63 ± 0.05* 0.42 ± 0.08 | *3.41 ± 0.76* 2.25 ± 0.68 | *3.90 ± 0.87* 2.69 ± 0.81 | *3.79 ± 0.85* 2.62 ± 0.79 | *12.6 %* 16.4 % |
| Ambient air 5 | Suburbs of Paris | 0.37 ± 0.03 3.4 ± 0.1 | *0.76 ± 0.08* 0.65 ± 0.12 | *2.72 ± 0.61* 2.54 ± 0.82 | *2.58 ± 0.58* 2.51 ± 0.81 | *2.77 ± 0.62* 2.61 ± 0.85 | *5.4 %* 2.7 % |
| Ambient air 6 | Suburbs of Paris | 0.37 ± 0.05 4.1 ± 1.0 | *0.62 ± 0.04* 0.46 ± 0.09 | *2.75 ± 0.50* 2.24 ± 0.60 | *2.78 ± 0.62* 2.96 ± 0.79 | *2.66 ± 0.59* 2.79 ± 0.75 | *19.1 %* 24.3 % |
| Ambient air 7 | Suburbs of Paris | 0.40 ± 0.01 4.7 ± 0.7 | *0.87 ± 0.05* 0.76 ± 0.08 | *3.85 ± 0.86* 1.86 ± 0.74 | *4.06 ± 0.91* 2.04 ± 0.69 | *4.01 ± 0.90* 2.02 ± 0.80 | *5.2 %* 8.8 % |
| Ambient air 8 | Suburbs of Paris | 0.42 ± 0.07 4.3 ± 0.7 | *0.78 ± 0.06* 0.71 ± 0.07 | *1.91 ± 0.43* 2.09 ± 0.61 | *2.22 ± 0.50* 2.53 ± 0.73 | *2.16 ± 0.48* 2.45 ± 0.72 | *14.0 %* 17.4 % |




**Table 3**. Mean and standard deviation multiple scattering correction $\overline{C_{ref}}$ at 450 and 660 nm for dust,
kaolinite, and ambient air. The $\overline{C_{ref}}$ was calculated as the mean of the $C_{ref}^{*}$, $C_{ref}$(W2003), and
$C_{ref}$(C2010) obtained at each wavelength for the different aerosol types. As a reminder: $C_{ref}^{*}$ is the
multiple scattering correction obtained not taking into account the shadowing effect correction in
aethalometer data; $C_{ref}$(W2003) and $C_{ref}$(C2010) take the shadowing effect correction into account, by
using the parametrisations by Weingartner et al. (2003) and Collaud Coen et al. (2010), respectively.

|  | $\overline{C_{ref}}$ | |
|---|---|---|
|  | **450 nm** | **660 nm** |
| Mineral dust | 2.09 ± 0.22 | 1.92 ± 0.17 |
| Kaolinite | 2.49 ± 0.02 | 2.32 ± 0.01 |
| Ambient air | 3.31 ± 0.75 | 2.32 ± 0.35 |



**Figure 1.** Experimental setup used for the aethalometer intercomparison experiments.

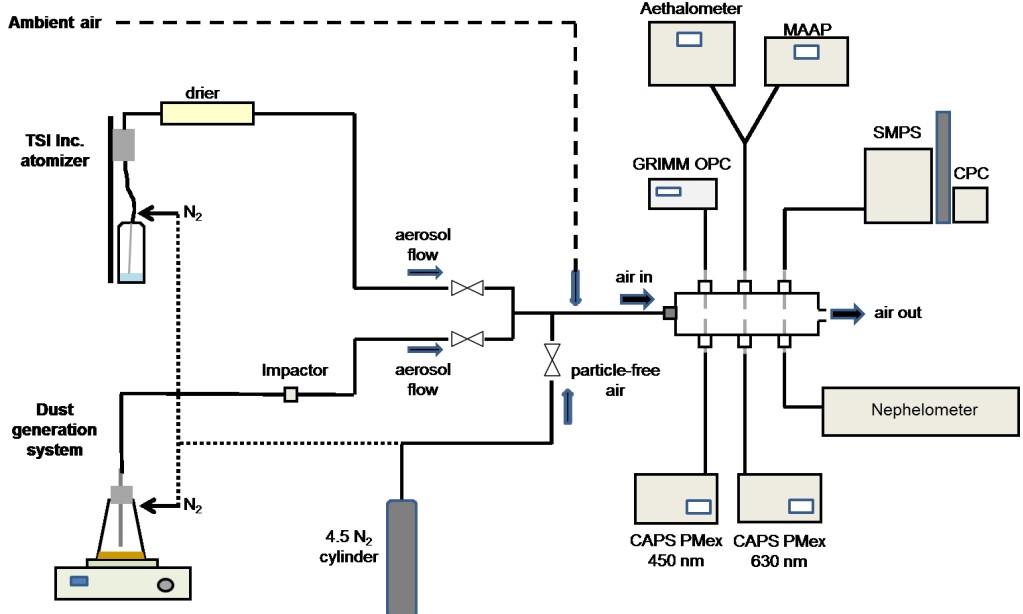








**Figure 2.** Temporal series of experiments showing the measured optical data at 660 nm. The different
panels show (from the top to the bottom): (i) the shadowing-corrected aethalometer attenuation at 660
nm (data corrected with the R formulation by Collaud Coen et al. (2010) (referred as R(C2010)) are
shown) and the MAAP aerosol absorption coefficient; (ii) the aerosol extinction at 660 nm
extrapolated from CAPS PMex measurements and estimated as the sum of nephelometer scattering
and MAAP absorption; (iii) the extinction aerosol Ångstrom exponent; (iv) the aerosol single scattering
albedo at 660 nm. Each point in the plot corresponds to 2 min average data. The x-axis indicates the
data point sequential number. Experiments with dust samples and kaolinite were realised between the
3$^{rd}$ and the 9$^{th}$ of November 2016 and lasted between 1 and 2 hours each. Ambient air data were
collected at different steps between the 8$^{th}$ and the 14$^{th}$ November 2016 for a total of 7 hours of
measurements.

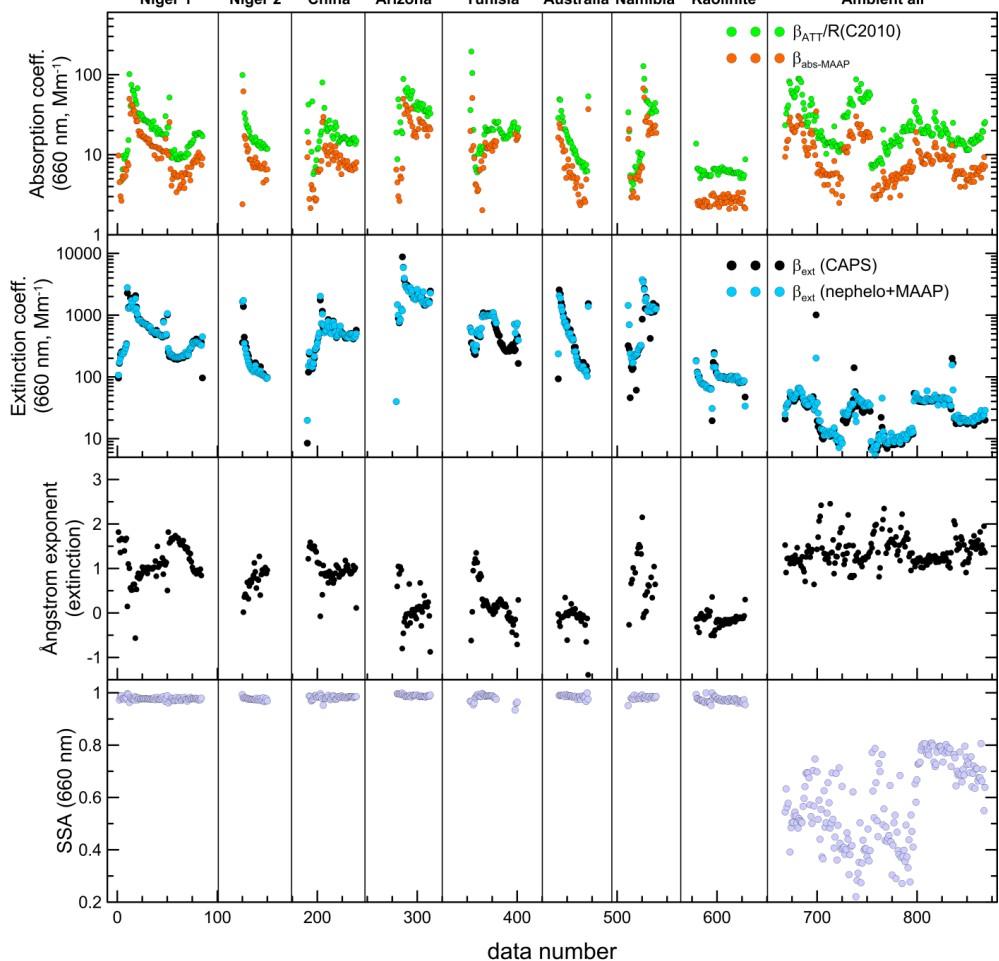






**Figure 3.** Ammonium sulfate experiment. Left panel: temporal evolution of the extinction and scattering coefficients measured by the CAPS PMex and the nephelometer at 450 nm (blue scale) and 630 nm (red scale). Each point in the plot corresponds to 2 min average data. Right panel: CAPS PMex versus nephelometer data (10 minutes averages). The y=x line and the results of the linear fit between CAPS and nephelometer data are also shown in the plot.

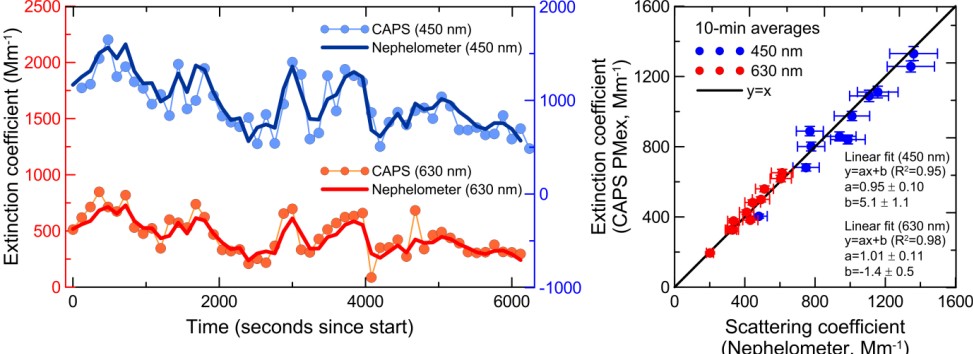



**Figure 4.** CAPS PMex extinction coefficient extrapolated at 660 nm versus nephelometer+MAAP
calculated extinction at 660 nm for all experiments (dust, kaolinite, ambient air). Each point in the plot
corresponds to 10 min average data. The y=x line and the results of the linear fit between CAPS and
nephelometer+MAAP data are also shown in the plot.

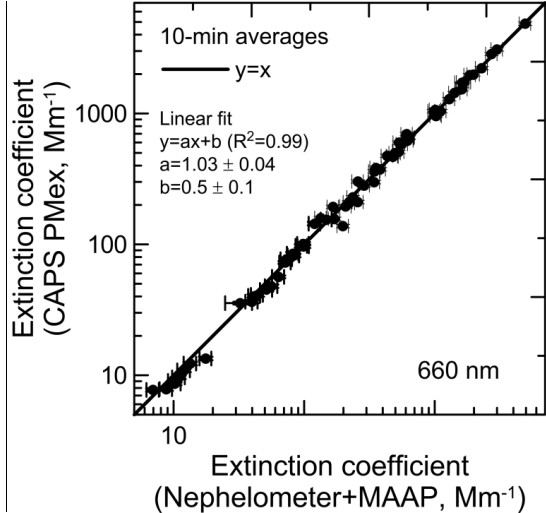


**Figure 5.** Left panel: estimated f values versus (1-SSA) at 660 nm for dust aerosols. Different
symbols are used to distinguish between dust from different sources. Right panel: f versus SSA at



660 nm for all experiments. Different symbols are used to distinguish between different aerosol types.
The results of the linear fit between f and (1-SSA) are also reported. Data from Weingartner et al.
(2003) (W2003) (extracted from their Figure 4) are also shown in the plot for comparison.



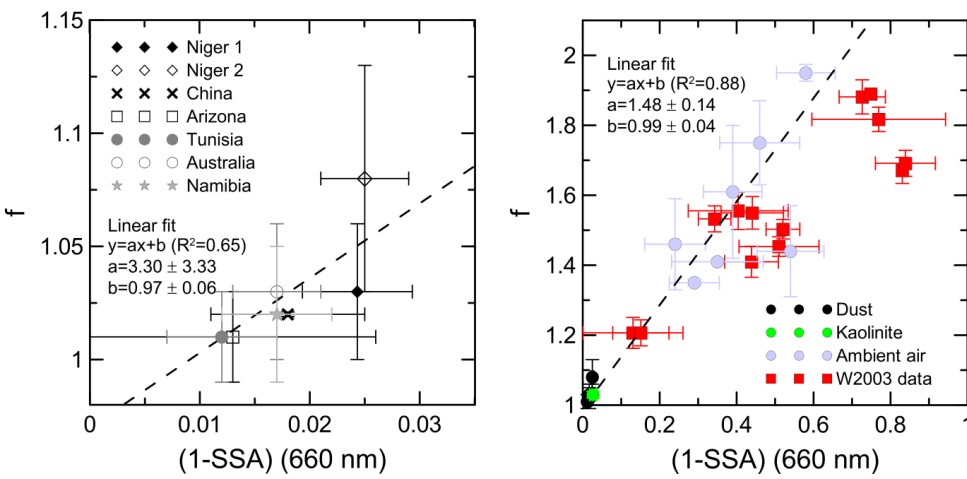







**Figure 6.** Top panel: $C_{ref}$(W2003) (multiple scattering correction obtained by taking into account the shadowing effect correction using the parametrisations by Weingartner et al. (2003)) versus SSA at 450 and 660 nm for mineral dust samples analysed in this study. Different symbols are used to distinguish between dust from different sources. As indicated in Table 2, the difference between $C_{ref}^{*}$, $C_{ref}$(W2003), and $C_{ref}$(C2010) is very low for mineral dust aerosols. Bottom panel: $C_{ref}$ versus SSA at 450 and 660 nm for the different aerosol samples analysed in this study. Different symbols are used to distinguish between different aerosol types. Data for both $C_{ref}$(W2003) and $C_{ref}^{*}$ (multiple scattering correction obtained not taking into account the shadowing effect correction in aethalometer data) are shown for ambient air aerosols, while for dust and kaolinite, for which the difference between the different formulations is very low, only $C_{ref}$(W2003) is reported. Data from Weigartner et al. (2003) (W2003) ($C_{ref}$ from their Table 3, and SSA extracted from their Fig. 4) and Collaud Coen et al. (2010) (C2010) (extracted from their Fig. 5) at 660 nm are also shown in the plot for comparison. The results of the linear fits between $C_{ref}$ and SSA for mineral dust and for the entire dataset are also shown in the plot.

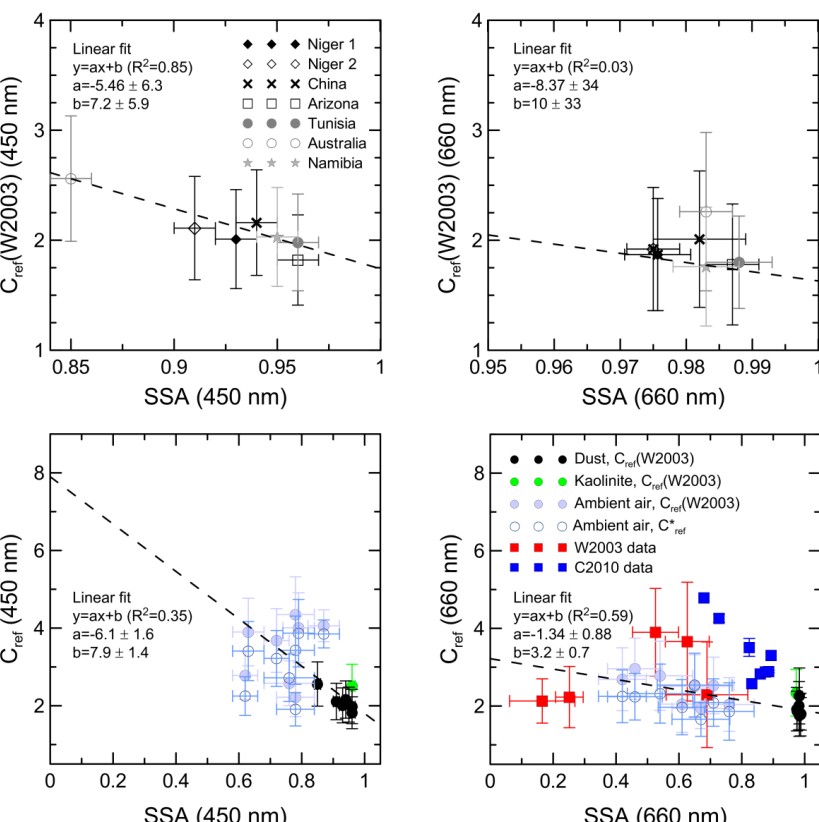



**Figure 7.** Examples of number size distribution (normalised to the total number concentration) for
ammonium sulfate, dust (Niger sample), kaolinite, and ambient air aerosols. Data refer to the mean
over each experiment as measured from the SMPS and the OPC. Error bars (standard deviations)
have been omitted for the sake of clarity.

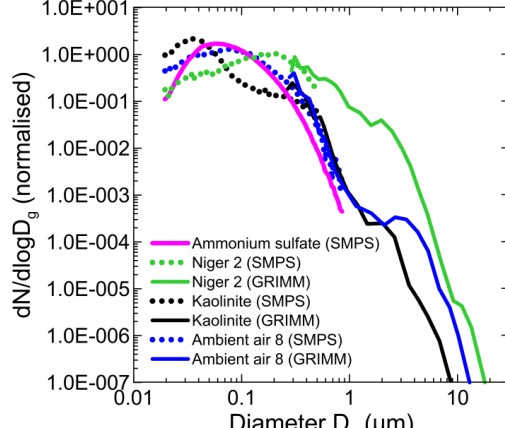







**Figure 8.** Top panel: $C_{ref}$(W2003) (multiple scattering correction obtained by taking into account the
shadowing effect correction using the parametrisations by Weingartner et al. (2003)) at 450 and 660
nm versus the effective diameter coarse $D_{eff,coarse}$. for mineral dust samples analysed in this study.
Different symbols are used to distinguish between dust from different sources. Bottom panel: $C_{ref}$ at
450 and 660 nm versus the effective diameter coarse $D_{eff,coarse}$ for the different aerosol samples
analysed in this study. Different symbols are used to distinguish between different aerosol types. Data
for both $C_{ref}$(W2003) and $C_{ref}^{*}$ (multiple scattering correction obtained not taking into account the
shadowing effect correction in aethalometer data) are shown for ambient air aerosols, while for dust
and kaolinite, for which the difference between the different formulations is very low, only $C_{ref}$(W2003)
is reported. The results of the linear fits between $C_{ref}$ and $D_{eff,coarse}$ for mineral dust and for the entire
dataset are also shown in the plot.

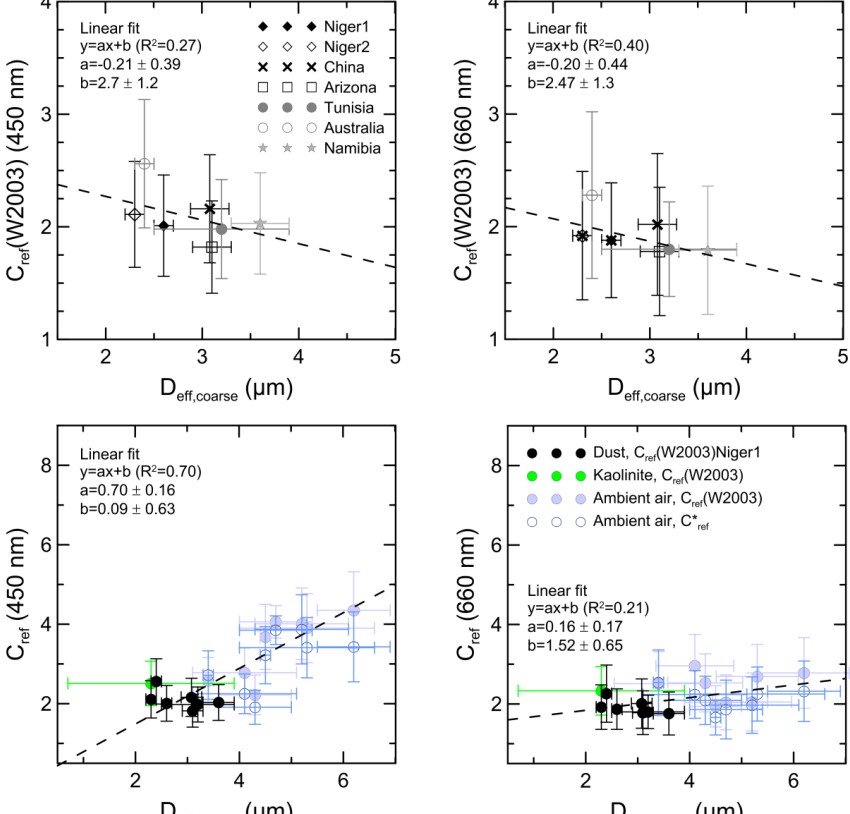
