# Peer review of "Aethalometer multiple scattering correction Cref for mineral dust aerosols"

_Atmospheric Measurement Techniques, 2017_

## Referee Comment (RC1) · Anonymous Referee #2 · 25 Apr 2017

This is a very nice paper – it's well-written and describes a well-designed experiment with useful results. The paper is appropriate for AMT. Kudos to the authors for making multiple checks/closure investigations on the measurements to make sure the data were consistent. I've made some minor editorial suggestions below. I guess some might also be considered science comments, but they are also minor.

Minor editorial and minor science comments:

Line 40 – Replace 'As for today,' with 'Currently'

Line 52 – Change to 'This is particularly true when compared to other aerosol species, such as soot, for which. . .'

Line 57 – should be '. . .global scales. . .'

Line 69-70 – Change to 'One instrument used to obtain aerosol light absorption. . .'

Line 72-73 – Change to 'The aethalometer reports equivalent black carbon mass concentration. . .' [comment: Petzold et al 2013 suggest the terminology 'equivalent black carbon' ]

Line 98 – Change to 'Thus, the value of . . .'

Line 119 – Change to 'The MAAP is commonly assumed to provide the most reliable filter-based, direct estimate. . .' [I think photoacoustic spectrometers are typically considered more reliable than filter based absorption measurements as there's no filter involved to confound the measurement]

Line 122 – Change to '. . .although Müller et al. (2011) measured. . .'

Line 123 – wavelength is spelled wrong

Line 134 – Move to Line 108 after the sentence 'The experimental set-up. . .' and change so it reads ' Instrumental details and uncertainties are summarized in Table 1.

Line 156-159 – say whether any conditioning (drying) was done to ambient particles.

Line 258 – change to '. . .which was then applied to extrapolate beta_sca to 630 and 660 nm.'

Line 285 – missing parenthesis around '13)' [comment – I'd probably call these equations 12a and 12b or just have one equation with d1 and d2 and say the range is 0.3-1.0 um for fine and 1-10 um for coarse.

Line 295++ – change to '. . .uncertainties of . . .' [Comment – I think it's more common/standard to refer to the 'uncertainty of' rather than the 'uncertainty on' so lots of instances to change in this paragraph

Line 317 – change to '. . .performance of the. . .'

[Figure]

Line 320 – change to 'This is further demonstrated by ...'

Line 316-324 –instrument uncertainties are listed in Table 1. This would be a good place to cite the instrument uncertainties and note that the difference is well within the uncertainties for the two instruments. Sherman et al (2015) supplemental materials is a good reference for the nephelometer uncertainties.

Line 351 – change to 'In contrast, for more absorbing...'

Line 361-362 – Probably should move this sentence into previous sentence rather than have a 1 sentence paragraph. In some ways it seems in conflict with the previous paragraph where you discuss Crefs being larger/smaller than each other depending on the ATT threshold. Can you make a plot or include numbers for the lower threshold in a table to definitively demonstrate that the 10 or 20% ATT threshold doesn't make a difference? Or maybe just put this sentence (lines 361-362) in the previous paragraph before the larger/smaller discussion so that the reader knows that, despite the Crefs being larger or smaller for the 10% versus the 20% threshold, the absolute difference is very small.

Line 364 - change to '...are reported...'

Line 372-381 - Were the ambient aerosol particles dried in any way? If not, does the ambient SSA vary with ambient and/or measurement RH? (I don't know, but am guessing Paris might be damp/humid in November). The TSI nephelometer tends to run warmer than many other instruments so potentially could have discrepancies in scattering estimate if neph measure of scattering drives off more water than CAPS-MAAP estimate of scattering. There's a slight suggestion of that in Fig 4 where I think the lowest group of extinction points are for ambient air and they look to be more below the 1:1 line than the other points (fig 4 is log scale, so hard to tell!). The closure still looks great and the focus of this paper was on lab generated dust so I'm more just curious.

Line 382 – change to '. . .serve two purposes.'

Line 386 – change to '. . .on relative amounts of particle absorption. . .'

Line 407-408 – could site Lack et al (2008) here – they saw enhanced absorption for filter-based measurements when more organic was present (for PSAP not aethalometer, but I imagine the there could be a similar effect).

Line 412-414 – it should be relatively straightforward (although admittedly annoying – sorry!) to recalculate results for MAAP at 630 nm to see how much of a role this wavelength discrepancy might play. I see from the acknowledgements that Andreas Petzold advised on this paper – perhaps ask him what he thinks about the MAAP measurement wavelength value.

Line 424 – change to '. . .particles, and may be linked. . .'

Line 426 – delete 'In correspondence,'

Line 435 – [comment – interesting that kaolinite has a significantly different absorption Angstrom exponent than dust. Isn't it often used as a surrogate for dust? Does this have any implications?

Line 442 – change to 'In contrast, no dependence of Cref on Deff,fine is found (R2<=0.44, not shown).'

Line 454++ change to 'Using these values of Cref, the dust absorption coefficient estimated by the aethalometer will be about 2% (450 nm)) and 11% (660 nm) higher than obtained. . .'

Line 474 – delete ', even if beyond the scope of the paper,'

Line 483-484 – change to 'This trend was only observed when the entire dataset was considered, but not if the dataset was limited to just the dust observations, making it difficult to draw clear conclusions.

Line 485 change '..of Cref is required...' Line 733 (and line 794) – change to '...(referred to as R(C2010))...'

Line 738 (and line 818) – change to '...kaolinite occurred between the...'

Table 1 – where do these uncertainty values come from? There are more recent (better!) references for the nephelometer uncertainty (e.g., Sherman et al 2015 – see their supplemental materials).

Figure 5 – why are f values so different for niger 1 and niger 2 and does this have an effect on results? Suggests results aren't totally reproducible.

Figure 8d (lower right) should the word 'niger 1' be in the figure legend? If so, there should be a space between it and (W2003)

General comment - A paper that might be of interest (if you haven't seen it) is Engelbrecht et al (2016) which has optical properties (e.g., SSA) for a bunch of different types of dust (i.e., dust from many different locations). I don't think you need to cite it (though you could). They used a photoacoustic instrument with a reciprocating nephelometer to obtain dust SSA values. If you and they have any overlapping dust samples it'd be nice to show/mention that the aethalometer had a similar response to dust as the photoacoustic since it's much simpler/cheaper to operate an aethalometer than a photoacoustic. Filter-based absorption instruments are often looked down on by some segments of the measurement community.

References

Engelbrecht et al (2016) Atmos. Chem. Phys, 16, 10809, 2016

Lack et al (2008) Aerosol Science and Technology, 42:1033–1041, 2008

Petzold et al (2013) Atmos. Chem. Phys., 13, 8365–8379, 2013

Sherman et al (2015) Atmos. Chem. Phys., 15, 12487–12517, 2015

---

## Referee Comment (RC2) · Anonymous Referee #3 · 28 Apr 2017

This is a very interesting manuscript of great importance for measuring mineral dust absorption coefficients with the AE31 aethalometer. It should be accepted for publication in AMT after the following comments have been taken into account.

1. Would these results be relevant to the currently sold AE33 aethalometer and would the authors expect the same Cref values?

2. L9-10: The abstract needs a better definition of Cref.

3. L83 and elsewhere: "Shadowing Effect": While this has a meaning in the geometric optics regime (x»1), it is completely meaningless for particle sizes comparable to or smaller then the wavelength. As this study encompasses both cases, a different expression (e.g., loading effect) should be used.

4. As three different kinds of aerosol absorption measurements (AE31, MAAP, difference method) form the core of this manuscript, general references on absorption measurements should be added such as the two major reviews of atmospheric and aerosol absorption by Horvath (1993) and Moosmuller et al. (2009).

5. L239-255: The Nephelometer truncation correction needs error estimations for both methods. Also were the particles sampled approximately spherical (SEM) images and what errors are expected from the assumption of spherical dust particles.

6. L374-375: "The wide range of [SSA] values indicates the occurrence of particles with very different absorption properties, henceforth chemical composition." It either indicates different chemical composition (or complex refractive index) and/or different size distribution as SSA strongly depends on both (e.g., Moosmuller and Arnott, 2009).

7. L415-427: When discussing particle size distributions, please always clarify if you are talking about number size or volume (mass) size distribution.

8. L457: "Given that the maximum intensity of the solar spectrum occurs at about 700 nm,…" I always thought that the maximum intensity (per wavelength interval) occurred around 500 nm. Please explain!

9. Some reference citations are inappropriate. For example, replace (Sokolik et al., 1999; L40) with (Sokolik and Toon, 1999; L40). Please check others!

10. P. 15-18: REFERENCES. This listing is incomplete and needs to be checked and completed! For example, Highwood and Ryder, 2014 (L38), Arnott et al., 2005 (L86), Petzold et al., 2004 (L114) are missing in the list of references.

REFERENCES

Horvath, H. (1993). "Atmospheric Light Absorption - A Review." Atmospheric Environment 27A(3): 293-317.

Moosmuller, H., R. K. Chakrabarty and W. P. Arnott (2009). "Aerosol Light Absorption

and its Measurement: A Review." Journal of Quantitative Spectroscopy & Radiative Transfer 110(11): 844-878.

Moosmuller, H. and W. P. Arnott (2009). Particle Optics in the Rayleigh Regime. J. Air & Waste Manage. Assoc., 59, 1028-1031.

---

## Referee Comment (RC3) · Anonymous Referee #1 · 9 May 2017

The authors have done a commendable job of executing a well-designed experiment to measure Cref for mineral dust aerosols. The experimental methods were carefully designed, with sufficient redundancy to test closure in the data. They applied the measurements to several aerosol types to determine the role of single scatter albedo and wavelength dependence on their values. The manuscript is well written and I recommend the paper be published after attending to minor comments below.

Comments Line 12: I suggest spelling out "CAPS PMex", and including "respectively" after "nephelometer".

Line 19: Change "The calculated mean Cref.." to "The calculated mean and one standard deviation Cref", or something along those lines so the reader knows what the numbers in parentheses refer to.

Line 21: Does the Cref=2.14 correspond to a specific wavelength? If so, include here.

Line 22: Does the 3% change correspond to both wavelengths?

Line 26: Include "respectively" after 660 nm.

Line 52: Include "such" between "species" and "as soot"

Line 54: The "-" in my version reads as a division sign, between ∼100-100000 and ∼0.1-100.

Line 99: Can the authors clarify as to what they mean by "optimized"?

Line 109: Correct "wavelentgth" to "wavelength"

Line 121: Same as previous comment.

Line 137: Please clarify sentence "so the same aerosol size distribution as input for all instruments". It seems to be missing a word.

Line 189: Should ln(ATN) in equations 6a and 6b be ln(ATT) ?

Line 276: Can the authors provide more detail regarding how this "conversion" was accomplished? Did they calibrate the OPC to provide a parameterization between refractive index and geometric and optical size? Can they comment on the role of relative humidity and how this might impact their data, since it didn't appear, especially in the ambient outdoor measurements, that they controlled RH? Addition of water would affect refractive index and change the instrument response.

Line 270: Please state the size range of the fine and coarse mode. It can be read off the integrals in equations 12 and 13 but would be clearer in the text.

Line 299: Were all of the Niger samples from size different areas combined to form 2 for the experiments?

Line 314: Why was the OPC not included in this control? (line 309-310).

Line 342: Change "251" to "2.51" (I assume this is a typo).

Line 380: Can the authors mention what the error bars refer to in this Figure and in the discussion for the following figures?

Line 465: What about the dependence of Cref and the coarse component at 450nm? (Figure 8, lower left).

---

## Author Comment (AC1) · 10 Jul 2017

**Revision of the paper: "Aethalometer multiple scattering correction $C_{ref}$ for mineral dust aerosols" by C. Di Biagio et al.**

**Answers to reviewers**

The authors wish to thank the reviewers for their valuable comments which helped to improve the quality and readability of the manuscript. Answers to the reviewer's comments are reported in the following (questions in black, answers in red).

**Referee #1**

The authors have done a commendable job of executing a well-designed experiment to measure Cref for mineral dust aerosols. The experimental methods were carefully designed, with sufficient redundancy to test closure in the data. They applied the measurements to several aerosol types to determine the role of single scatter albedo and wavelength dependence on their values. The manuscript is well written and I recommend the paper be published after attending to minor comments below.

Line 12: I suggest spelling out "CAPS PMex", and including "respectively" after "nephelometer".
The text was changed accordingly.

Line 19: Change "The calculated mean Cref.." to "The calculated mean and one standard deviation Cref", or something along those lines so the reader knows what the numbers in parentheses refer to.
The text was changed accordingly.

Line 21: Does the Cref=2.14 correspond to a specific wavelength? If so, include here.
This has been rewritten as: "..higher than that obtained by using $C_{ref}$=2.14 at both 450 and 660 nm, as usually assumed in the literature".

Line 22: Does the 3% change correspond to both wavelengths?
The change corresponds to the 660 nm wavelength. This is now specified in the abstract.

Line 26: Include "respectively" after 660 nm.
The text was changed accordingly.

Line 52: Include "such" between "species" and "as soot"
The text was changed accordingly.

Line 54: The "-" in my version reads as a division sign, between~100-100000 and ~0.1-100.
The text was changed accordingly.

Line 99: Can the authors clarify as to what they mean by "optimized"?
The sentence was rewritten as: "Henceforth, in this work we present the experimental estimate of $C_{ref}$ for mineral dust aerosols at 450 and 660 nm obtained from a laboratory-based intercomparison study."

Line 109: Correct "wavelentgth" to "wavelength"
The text was changed accordingly.
Line 121: Same as previous comment.
The text was changed accordingly.

Line 137: Please clarify sentence "so the same aerosol size distribution as input for all instruments". It seems to be missing a word.
The sentence was rewritten as: "Their length, varying between 0.3 and 0.7 m, was adjusted based on the flowrate of each instrument to ensure an equivalent particle loss, so that the same aerosol size distribution is in input to the different instruments."

Line 189: Should ln(ATN) in equations 6a and 6b be ln(ATT) ?
The text was changed accordingly.

Line 276: Can the authors provide more detail regarding how this "conversion" was accomplished? Did they calibrate the OPC to provide a parameterization between refractive index and geometric and optical size? Can they comment on the role of relative humidity and how this might impact their data, since it didn't appear, especially in the ambient outdoor measurements, that they controlled RH? Addition of water would affect refractive index and change the instrument response.
This part was rewritten as: "The OPC optical-equivalent nominal diameters were converted into sphere-equivalent geometrical diameters ($D_g$) by taking into account the aerosol complex refractive index. This consisted in recalculating the OPC calibration curve for different complex refractive index values. For dust aerosols the refractive index was varied in the range 1.47-1.53 (n) and 0.001-0.005$i$ (k) following the literature (see Di Biagio et al., 2017) and $D_g$ was set at the mean ± one standard deviation of the values obtained for the different n and k. For kaolinite the OPC diameter conversion was performed by setting the refractive index at 1.56-0.001$i$. For ambient air the refractive index was set at 1.60-0.01$i$, a value that represents a medium absorbing urban polluted aerosol (see Di Biagio et al., 2016). The impact of humidity on the refractive index of ambient aerosols and associated changes OPC response are not taken into account. The relative humidity was always below 35% during ambient air measurements, which implies a very small particle growth."

Line 270: Please state the size range of the fine and coarse mode. It can be read off the integrals in equations 12 and 13 but would be clearer in the text.
The size ranges of the fine and coarse modes are now explicitly stated in the main text.

Line 299: Were all of the Niger samples from size different areas combined to form 2 for the experiments?
The two Niger samples, as reported in Table 2, correspond to the same soil sample collected at the rural area of Banizoumbou. We decided to duplicate the experiments for the Niger soil in order to test the repeatability of the results.

Line 314: Why was the OPC not included in this control? (line 309-310).
The main objective of the control experiment was to verify the performance of optical instruments. We did not consider in this case necessary to have a redundancy also on size distribution data.

Line 342: Change "251" to "2.51" (I assume this is a typo).
The text was changed accordingly.

Line 380: Can the authors mention what the error bars refer to in this Figure and in the discussion for the following figures?
The captions of Fig. 5, 6, and 8 have been modified to include an explanation of what error bars refer to.

Line 465: What about the dependence of Cref and the coarse component at 450nm? (Figure 8, lower left).
The dependence found for $C_{ref}$ at 450 nm against $D_{eff,coarse}$ can be related to the fact that when large absorbing ambient aerosols deposit on the filter the scattering from the filter fibres can increase due to some multiple scattering with these particles. However, this is just a hypothesis and a more detailed investigation on this topic should be addressed to clarify this behaviour. Given that the main focus of the paper is on mineral dust we decided not to comment this result in the paper.

**Referee #2**
This is a very nice paper – it's well-written and describes a well-designed experiment with useful results. The paper is appropriate for AMT. Kudos to the authors for making multiple checks/closure investigations on the measurements to make sure the data were consistent. I've made some minor editorial suggestions below. I guess some might also be considered science comments, but they are also minor.

Minor editorial and minor science comments:

Line 40 – Replace 'As for today,' with 'Currently'
The text was changed accordingly.

Line 52 – Change to 'This is particularly true when compared to other aerosol species, such as soot, for which...'
The text was changed accordingly.

Line 57 – should be '...global scales...'
The text was changed accordingly.

Line 69-70 – Change to 'One instrument used to obtain aerosol light absorption...'
The text was changed accordingly.

Line 72-73 – Change to 'The aethalometer reports equivalent black carbon mass Concentration ...' [comment: Petzold et al 2013 suggest the terminology 'equivalent black carbon' ]
The text was changed accordingly.

Line 98 – Change to 'Thus, the value of...'
The text was changed accordingly.

Line 119 – Change to 'The MAAP is commonly assumed to provide the most reliable filter-based, direct estimate...' [I think photoacoustic spectrometers are typically considered more reliable than filter based absorption measurements as there's no filter involved to confound the measurement]
The text was changed accordingly.

Line 122 – Change to '...although Müller et al. (2011) measured...'
The text was changed accordingly.

Line 123 – wavelength is spelled wrong
The text was changed accordingly.

Line 134 – Move to Line 108 after the sentence 'The experimental set-up...' and change so it reads ' Instrumental details and uncertainties are summarized in Table 1.
The text was changed accordingly.

Line 156-159 – say whether any conditioning (drying) was done to ambient particles.
The sentence was changed in: "-ambient pollution aerosols were sampled by opening the manifold to the exterior ambient air. Ambient aerosols were not dried."

Line 258 – change to '...which was then applied to extrapolate beta_sca to 630 and 660 nm.'
The text was changed accordingly.

Line 285 – missing parenthesis around '13)' [comment – I'd probably call these equations 12a and 12b or just have one equation with d1 and d2 and say the range is 0.3-1.0 um for fine and 1-10 um for coarse.
The text was changed accordingly.

Line 295++ – change to '...uncertainties of...' [Comment – I think it's more common/standard to refer to the 'uncertainty of' rather than the 'uncertainty on' so lots of instances to change in this paragraph
The text was changed accordingly.

Line 317 – change to '...performance of the...'
The text was changed accordingly.

Line 320 – change to 'This is further demonstrated by...'
The text was changed accordingly.

Line 316-324 –instrument uncertainties are listed in Table 1. This would be a good place to cite the instrument uncertainties and note that the difference is well within the uncertainties for the two instruments. Sherman et al (2015) supplemental materials is a good reference for the nephelometer uncertainties.
Part of the paragraph has been rewritten as: "As expected for this purely scattering aerosol (Toon et al., 1976), the nephelometer scattering and the CAPS extinction at 450 and 630 nm were in very good agreement (less than 4% difference) during the whole duration of the experiment. This is well below the single instrument uncertainty of ±9% for the nephelometer (Sherman et al., 2015) and ±5% for the CAPS (Massoli et al., 2010)."

Line 351 – change to 'In contrast, for more absorbing...'
The text was changed accordingly.

Line 361-362 – Probably should move this sentence into previous sentence rather than have a 1 sentence paragraph. In some ways it seems in conflict with the previous paragraph where you discuss Crefs being larger/smaller than each other depending on the ATT threshold. Can you make a plot or include numbers for the lower threshold in a table to definitively demonstrate that the 10 or 20% ATT threshold doesn't make a difference? Or maybe just put this sentence (lines 361-362) in the previous paragraph before the larger/smaller discussion so that the reader knows that, despite the Crefs being larger or smaller for the 10% versus the 20% threshold, the absolute difference is very small.
The paragraph has been rewritten as: "Differences within 2.8% were obtained between $C_{ref}^{*}$, $C_{ref}$(W2003) and $C_{ref}$(C2010) at 450 and 660 nm for weakly-absorbing dust and kaolinite. In contrast, for more absorbing ambient air aerosols the differences between $C_{ref}^{*}$, $C_{ref}$(W2003) and $C_{ref}$(C2010) were in the range 2.7% to 24.3%. The different ATT threshold assumed here (20%) compared to W2003 and C2010 (10%) has a negligible impact (less than 1% difference) on the results. In some cases (ambient air 1–2 and Niger 1 samples), however, we obtained $C_{ref}$(C2010)>$C_{ref}$(W2003); these cases correspond to a mean aethalometer measured ATT<10%, for which R(W2003)>R(C2010), and this explains the larger $C_{ref}$(C2010). Conversely, $C_{ref}$(C2010)<$C_{ref}$(W2003) when the measured ATT was ~15-20%, yielding R(W2003)<R(C2010). The percent difference between the obtained $C_{ref}$(W2003) and $C_{ref}$(C2010) increased for decreasing SSA due to the increase of the R(W2003) to R(C2010) absolute difference for decreasing SSA. When averaging data for all ambient air samples, the two formulations yield very similar values. For example, at 660 nm the mean $C_{ref}$(W2003) was 2.44 (± 0.38), less than 2% larger than the mean $C_{ref}$(C2010) of 2.39 (± 0.35)."

Line 364 - change to '...are reported...'
The text was changed accordingly.

Line 372-381 - Were the ambient aerosol particles dried in any way? If not, does the ambient SSA vary with ambient and/or measurement RH? (I don't know, but am guessing Paris might be damp/humid in November). The TSI nephelometer tends to run warmer than many other instruments so potentially could have discrepancies in scattering estimate if neph measure of scattering drives off more water than CAPS-MAAP estimate of scattering. There's a slight suggestion of that in Fig 4 where I think the lowest group of extinction points are for ambient air and they look to be more below the 1:1 line than the other points (fig 4 is log scale, so hard to tell!). The closure still looks great and the focus of this paper was on lab generated dust so I'm more just curious.
Ambient air is not dried (this is now explicitly stated in the main text). The nephelometer RH during ambient air measurements was between 20-35%, against the <15% RH during kaolinite, dust, and ammonium sulfate experiments. As discussed by the reviewer, the possible difference in RH conditions between the three optical instruments (nephelometer expected to have larger RH compared to the CAPS and the aethalometer) seems not to affect our data as the closure is always very good. Concerning the few points in Fig.4 with very small extinctions (less than 20 Mm-1) the nephelometer+aethalometer extinction was slightly larger than the CAPS (less than 10% difference, which is within the instrument's uncertainties). Any possible effect of RH is however difficult to investigate due to the limited RH range in our measurements.

Line 382 – change to '...serve two purposes.'
The text was changed accordingly.

Line 386 – change to '...on relative amounts of particle absorption...'
The text was changed accordingly.

Line 407-408 – could site Lack et al (2008) here – they saw enhanced absorption for filter-based measurements when more organic was present (for PSAP not aethalometer, but I imagine the there could be a similar effect).
The suggested reference was added to the text.

Line 412-414 – it should be relatively straightforward (although admittedly annoying– sorry!) to recalculate results for MAAP at 630 nm to see how much of a role this wavelength discrepancy might play. I see from the acknowledgements that Andreas Petzold advised on this paper – perhaps ask him what he thinks about the MAAP measurement wavelength value.
We evaluated the impact of the exact wavelength on the retrieved $C_{ref}$ by assuming 637 nm as the nominal MAAP wavelength and by using this value to extrapolate the absorption coefficient at 660 nm. We then used this new value to estimate $C_{ref}$. As expected, using 637 nm determines an increase in $C_{ref}$ at 660 nm. This increase is +8–14% for mineral dust, +3% for kaolinite, and +3–15% for ambient air aerosols, independently of the used formulation for $C_{ref}$ calculation (C2010, W2003, or $C_{ref}^{*}$).
In order to add this information in the main text we added the following text:
Sect. 2: "An estimate of the change in the obtained $C_{ref}$ due to the change in MAAP nominal wavelength from 670 to 637 nm is reported in Sect. 4.2;"
Sect. 4.2: "If the wavelength of 637 nm is assumed for the MAAP instead of 670 nm, as suggested by Müller et al. (2011), the average $C_{ref}$ at 660 nm would increase by up to ~15% for dust and ambient air (2.17±0.19 and 2.48±0.41, respectively) and ~3% for kaolinite (2.40±0.02)."

Line 424 – change to '...particles, and may be linked...'
The text was changed accordingly.

Line 426 – delete 'In correspondence,'
The text was changed accordingly.

Line 435 – [comment – interesting that kaolinite has a significantly different absorption Angstrom exponent than dust. Isn't it often used as a surrogate for dust? Does this have any implications?
First; we found some errors in the text and numbers between lines 432-436, which we rewrote as: "The $\alpha_E$ (shown in Fig.2) was ~0 for kaolinite, varied between about 0 and 2 for mineral dust aerosols, and between 0.5 and 2.5 for ambient air, indicating particles with variable sizes, both the sub-micron and the super-micron fractions. The absorption Ångström coefficient $\alpha_A$ obtained from aethalometer data was between 2.2 and 4 for dust, between 1 and 1.5 for kaolinite and between 0.5 and 1.5 for ambient air aerosols. "

To answer to your comment: yes, kaolinite is usually used as a surrogate of dust and this may lead to large uncertainties due to the differences in the size distribution and the composition between the two, which affect their absorbing behaviour, as we can see here for example in relation to the absorption Ångström coefficient $\alpha_A$. We decided however not to stress this point in the manuscript since any comment or conclusion should deserve a more systematic study, which was not the case for this paper.

Line 442 – change to 'In contrast, no dependence of Cref on Deff,fine is found (R2<=0.44, not shown).'
The text was changed accordingly.

Line 454++ change to 'Using these values of Cref, the dust absorption coefficient estimated by the aethalometer will be about 2% (450 nm)) and 11% (660 nm) higher than obtained...'
The text was changed accordingly.

Line 474 – delete ', even if beyond the scope of the paper,'
The text was changed accordingly.

Line 483-484 – change to 'This trend was only observed when the entire dataset was considered, but not if the dataset was limited to just the dust observations, making it difficult to draw clear conclusions.
The text was changed accordingly.

Line 485 change '..of Cref is required...'
The text was changed accordingly.

Line 733 (and line 794) – change to'...(referred to as R(C2010))...'
The text was changed accordingly.

Line 738 (and line 818) – change to '...kaolinite occurred between the...'
The text was changed accordingly.

Table 1 – where do these uncertainty values come from? There are more recent (better!) references for the nephelometer uncertainty (e.g., Sherman et al 2015 – see their supplemental materials).
The reference by Sherman was added in Table 1 together with their estimated uncertainty of ~9% on the nephelometer scattering coefficient.

Figure 5 – why are f values so different for niger 1 and niger 2 and does this have an effect on results? Suggests results aren't totally reproducible.
The estimate of f values for Niger 1 and Niger 2 was 1.03 and 1.08 respectively, which corresponds to ~5% change. Even if not perfect, we consider these values sufficiently in accordance to prove the reproducibility of the results.

Figure 8d (lower right) should the word 'niger 1' be in the figure legend? If so, there should be a space between it and (W2003)
The plot was corrected.

General comment - A paper that might be of interest (if you haven't seen it) is Engelbrecht et al (2016) which has optical properties (e.g., SSA) for a bunch of different types of dust (i.e., dust from many different locations). I don't think you need to cite it (though you could). They used a photoacoustic instrument with a reciprocating nephelometer to obtain dust SSA values. If you and they have any overlapping dust samples it'd be nice to show/mention that the aethalometer had a similar response to dust as the photoacoustic since it's much simpler/cheaper to operate an aethalometer than a photoacoustic. Filter-based absorption instruments are often looked down on by some segments of the measurement community.
Only for China, Arizona, and Australia we found overlapping dust samples between our study and Engelbrecht et al. (2016). For these samples the comparison was quite good, despite the different wavelengths used in the two analyses. We included this comparison in the paper, with the following lines in Sect. 4.3: "In particular, our results for China, Arizona, and Australia samples are in line with published values by Engelbrecht et al. (2016), who used a photoacusting instrument to measure absorption of re–suspended dust aerosols. This would suggest the similar performances of the aethalometer compared to the photoacoustic technique. The SSA for kaolinite was 0.96–0.97 at 450 and 660 nm, in agreement with Utry et al. (2017) also using a photoacusting mehod to measure absorption (0.97 and 0.99 (±0.04) at 450 and 635 nm, respectively)."

[revised manuscript text omitted]

---

## Author Comment (AC3) · 10 Jul 2017

The comment was uploaded in the form of a supplement:
https://www.atmos-meas-tech-discuss.net/amt-2017-65/amt-2017-65-AC3-supplement.pdf